# BiRQA: Bidirectional Robust Quality Assessment for Images

Aleksandr Gushchin [1 2]   Dmitriy Vatolin [3]   Anastasia Antsiferova [1 2]

## Abstract

Full-Reference image quality assessment (FR IQA) is important for image compression, restoration and generative modeling, yet current neural metrics remain slow and vulnerable to adversarial perturbations. We present BiRQA, a compact FR IQA metric model that processes four fast complementary features within a bidirectional multiscale pyramid. A bottom-up attention module injects fine-scale cues into coarse levels through an uncertainty-aware gate, while a top-down cross-gating block routes semantic context back to high resolution. To enhance robustness, we introduce Anchored Adversarial Training, a theoretically grounded strategy that uses clean "anchor" samples and a ranking loss to bound pointwise prediction error under attacks. On five public FR IQA benchmarks BiRQA outperforms or matches the previous state of the art (SOTA) while running $\sim 3\times$ faster than previous SOTA models. Under unseen white-box attacks it lifts SROCC from 0.30-0.57 to 0.60-0.84 on KADID-10k, demonstrating substantial robustness gains. To our knowledge, BiRQA is the only FR IQA model combining competitive accuracy with real-time throughput and strong adversarial resilience.

## 1. Introduction

Image Quality Assessment (IQA) is a fundamental problem in computer vision with applications in image restoration, compression, and generative modeling. Full-Reference (FR) IQA estimates perceived quality by comparing a distorted image with its pristine reference. While classical approaches like PSNR and SSIM are fast, they overlook many complex perceptual details, driving interest in deep learning approaches. Yet even strong neural IQA models still face two pressing issues: (i) slow inference speed that limits real-time use, and (ii) high vulnerability to adversarial perturbations, threatening reliability in safety-critical applications.

Adversarial attacks introduce imperceptible perturbations that mislead NN-based IQA models. Despite recent defenses for FR-IQA, robustness benchmarks show that many metrics remain vulnerable (Gushchin et al., 2024), making them unsuitable for domains such as medical imaging, autonomous driving, and content authentication, where scores must remain trustworthy under both adversarial and benign perturbations. Moreover, these vulnerabilities allow attackers to manipulate image search results, as search engines rely on IQA metrics for ranking. Such attacks can also falsify public benchmark results (Huang et al., 2024; Wu et al., 2024) by exploiting weaknesses in IQA models and artificially boosting perceived algorithm quality. For example, incorporating a vulnerable IQA metric as a loss function in image restoration can degrade actual image quality (Ding et al., 2021) or cause visual artifacts (Kashkarov et al., 2024). These risks highlight the urgent need for an FR IQA method that combines accuracy with adversarial robustness.

In this work, we present **BiRQA**: a precise, fast, compact, and attack-resilient FR IQA metric. The model builds a multiscale feature pyramid and injects feature maps that capture important patterns for human visual system (gradient structure, color dissimilarities, and local binary patterns) into a lightweight neural network. Information flows bidirectionally, which is generally a novel concept in IQA: a bottom-up attention lifts fine artifacts with an uncertainty-aware gate that outputs strength and confidence, reducing error propagation across scales. Then a top-down cross-gating supplies global context. This flow reduces scale-specific blind spots, yielding more precise quality scores across unseen distortions. A reliability-aware aggregation head pools each scale with GeM and combines per-scale contributions via softmax-normalized confidence weights, producing an interpretable convex combination. Reliability is strengthened through anchor-based adversarial training (AT) that fine-tunes the model to preserve the ranking of adversarial predictions with respect to clean anchors. Theoretical analysis links the anchor-based optimization objective to a maximal pointwise

---

[1]Trusted AI Research Center RAS, Moscow, Russia [2]Lomonosov Moscow State University, Moscow, Russia [3]MSU Institute for Artificial Intelligence, Moscow, Russia. Correspondence to: Aleksandr Gushchin <alexander.gushchin@graphics.cs.msu.ru>, Anastasia Antsiferova <aantsiferova@graphics.cs.msu.ru>.

prediction error. Extensive experiments on standard IQA benchmarks show that BiRQA achieves superior accuracy while maintaining computational efficiency ($\sim$15 FPS on $1920 \times 1080$ images). Furthermore, our method generalizes across diverse distortion types and remains resilient to adversarial perturbations, outperforming existing methods in attack scenarios. The key contributions are:

- A novel FR IQA model architecture **BiRQA** uses bidirectional, uncertainty-aware cross-scale fusion with interpretable aggregation. The proposed CSRAM (fine→coarse) and SCGB (coarse→fine) modules exchange signals through learned gates, while a lightweight head aggregates scales with uncertainty-aware weights. The code will be publicly available.

- Theoretically grounded anchored AT uses clean anchor samples and a ranking loss to tighten a prediction error bound, boosting SROCC under attacks by up to 0.30 over the undefended model and 0.05 over the prior defenses.

- Extensive experiments on five public FR IQA benchmarks and four unseen white-box attacks show that BiRQA matches or surpasses previous SOTA metrics, runs $\sim 3\times$ faster than transformer methods, and improves integral robustness scores by up to 12%.

## 2. Related Work

**Full-Reference Image Quality Assessment.** Assessing the quality of images is critical for numerous applications, such as compression, super-resolution, and other image processing techniques. FR IQA methods evaluate perceptual quality by comparing a distorted image to its pristine reference. While PSNR is fast, it correlates poorly with the Human Visual System (HVS). Metrics like SSIM (Wang et al., 2004), MS-SSIM (Wang et al., 2003), FSIM (Zhang et al., 2011), SR-SIM (Zhang & Li, 2012), and VIF (Sheikh & Bovik, 2006) model structure, phase, saliency, or visibility to better match perception at low cost. These methods depend on heuristics and analytic features, ensuring computational efficiency. Deep learning further improves performance with models like LPIPS (Zhang et al. 2018) and DISTS (Ding et al. 2020). Transformers extend this via SwinIQA (Liu et al. 2022), IQT (Cheon et al. 2021), AHIQ (Lao et al. 2022), and the current state-of-the-art (SOTA) method TOPIQ (Chen et al. 2024), which adopts a multiscale top-down scheme with Cross-Self Attention. Several recent models further exploit multiscale attention. SwinIQA and IQT both address cross-scale information flow: SwinIQA relies on a heavy transformer pipeline, whereas proposed BiRQA model integrates lightweight CNN layers with perceptually grounded analytic features to retain speed. IQT

passes representations from coarse-to-fine layers, while BiRQA's Cross-Scale Residual Attention Module (CSRAM) exchanges information bottom-up, yielding a bidirectional interaction that improves detail recovery. Beyond multiscale attention, some works explore learning to rank image quality. RankIQA (Liu et al. 2017) trains a Siamese network to predict quality for NR IQA. To our knowledge, ranking has not yet been combined with adversarial training in FR IQA, a gap we address through the anchored ranking loss in BiRQA.

**Robust IQA Methods.** In FR IQA, an attack adds an imperceptible perturbation that deceives the metric. Attacks are classified as white-box, where gradients and parameters are known, or black-box, where only output scores are available (Chakraborty et al. 2021). IQA-specific attacks have been proposed in Korhonen & You 2022; Shumitskaya 2024; 2023; Zhang et al. 2022.

Defenses divide into certified and empirical categories. While certified defenses offer provable guarantees, they remain too slow for real-time use. Empirical strategies are more practical and often revolve around adversarial training or input purification. There are some works focused on NR-IQA, including input purifications (E-LPIPS, Kettunen et al. 2019), adversarial training (R-LPIPS (Ghazanfari et al. 2023), AT (Chistyakova et al. 2024)) and architecture modification (Grad.Norm., Liu et al. 2024). Although previous AT methods employ data augmentation or gradient regularization, our anchored adversarial training integrates a ranking loss to enhance robustness.

## 3. Method

FR IQA requires four properties still unmet by many deep learning models: (1) *accuracy*, (2) *low latency*, (3) *resilience to adversarial perturbations*, and (4) *sensitivity to multi-scale artifacts*. To meet these goals, we present BiRQA, a compact hybrid network that injects lightweight, human-interpretable feature maps into a bidirectional attention pyramid guided by HVS principles. Figure 1 sketches the pipeline: feature maps are arranged in a four-level pyramid, preserving fine detail at high resolution and summarizing global structure at lower resolutions. At each scale, an *AdaptiveFusion* block reweights channels, a bottom-up *Cross-Scale Residual Attention Module* (CSRAM) lifts fine-scale cues to coarser levels, and a top-down *Spatial Cross-Gating Block* (SCGB) feeds semantic context back to higher resolutions, completing the bidirectional exchange. A Reliability-Aware Head (RAH) pools per-scale representations and aggregates them via normalized reliability weights to produce the final score. Training minimizes a PLCC-oriented regression loss together with an anchored ranking loss.

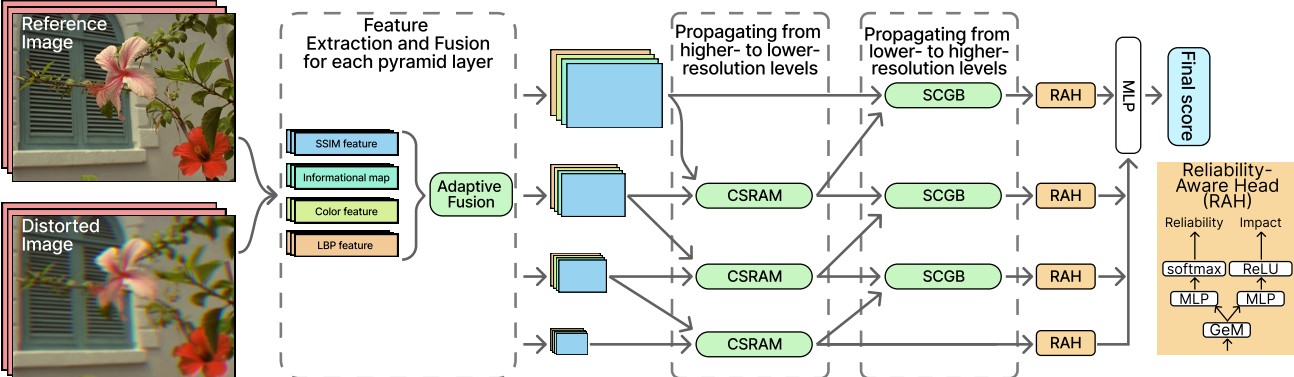

*Figure 1.* Overall scheme of BiRQA. A reference–distorted pair yields four feature maps per pyramid level. Cross-Scale Residual Attention Module (CSRAM) and Spatial Cross-Gating Block (SCGB) allow the model to pass information in both directions between scales. A Reliability-Aware Head (GeM + dual MLPs) estimates per-level impact and reliability.

## 3.1. Feature Extraction

Feature selection was guided by two primary criteria: computational efficiency and the ability to detect complementary types of image degradation. To keep model fast we explored various lightweight analytic features rather than building complex image representations from raw images. We evaluated 11 candidate features, including Gabor filters, wavelet transform, entropy map, edge map, detail loss measure (Li et al. 2011), VIF, and saliency. A total of 300+ feature combinations were tested. The following four features delivered the best accuracy–runtime trade-off; adding raw images as additional inputs to BiRQA worsened results. Full details are provided in Appendix B. The four chosen features address distinct aspects of quality degradation: (1) **SSIM map**: costs little to compute and provides a spatial implementation of the structural-similarity idea, which is consistent with how people compare distortions. (2) **Local informational content**: measures the variance of pixel intensities to estimate whether a region is highly informative. The selection of this feature was inspired by IW-SSIM (Wang & Li 2010). (3) **YCbCr color difference map**: isolates chroma shifts and color bleeding in channels aligned with the HVS. (4) **Local Binary Patterns (LBP)**: compares each pixel with its neighboring pixels to encode local texture information into binary patterns. This method has proven effective under adversarial attacks (Asmitha et al. 2024).

Unlike many recent IQA models that crop images to lower resolutions for faster computation and compatibility with pre-trained backbones, our approach avoids cropping to preserve global context and degradation-specific regions. Instead, we compute feature maps at the original resolution and integrate them into a pyramidal framework. This method uses four pyramid levels, each downscaled by a factor of two from the previous level, enabling the network to capture and aggregate multiscale degradation information effectively.

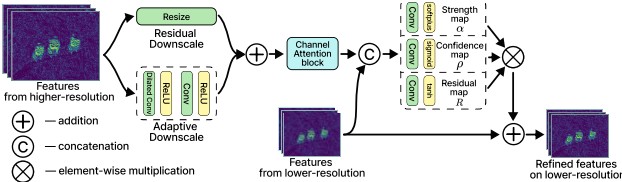

*Figure 2.* Scheme of the Cross-Scale Residual Attention Module (CSRAM) that lifts high-resolution cues to the next scale and injects them via uncertainty-aware gated residuals (strength $\alpha$, confidence $\rho$, and residual $R$) to refine the lower-resolution features.

## 3.2. BiRQA Network

Following feature extraction, the BiRQA network processes these features to compute the final quality score. Each feature map at every pyramid level is first preprocessed individually to highlight significant spatial regions. These preprocessed feature tensors we denote as $\{F_i^j\}, F_i^j \in \mathbb{R}^{D_j \times H_i \times W_i}$, where $H_i, W_i$ are feature map dimensions on $i$-th pyramid level, $D_j$ denoting the number of channels for feature $j$.

**Multi-feature Adaptive Fusion**. At pyramid level $i$, we concatenate the four features, compute a joint attention vector $\alpha_i^j = \sigma(\text{MLP}_i^j(\text{GAP}(F_i^0 \oplus F_i^1 \oplus F_i^2 \oplus F_i^3)))$, and obtain the fused tensor $G_i = \phi_i(\bigoplus_{j=0}^3 \alpha_i^j \odot F_i^j)$, where GAP is global average pooling, $\odot$ – element-wise multiplication and $\oplus$ – concatenation, $\phi_i$ – convolution. A Squeeze-and-Excitation block (Hu et al. 2018) adaptively recalibrates $G_i$ by "squeezing" spatial information into a channel descriptor via global pooling and then "exciting" (reweighting) each channel through a learned gating mechanism. This allows BiRQA to dynamically emphasize relevant feature channels and suppress noisy or redundant ones, improving robustness at negligible extra cost.

Naive resizing can blur high-frequency artifacts and propagate false positives; we transmit a *gated residual* whose

*strength* and *confidence* are learned separately. We introduce **Cross-Scale Residual Attention Module** (**CSRAM**, Fig. 2) to interconnect scales. Fine-scale artifacts arise first; CSRAM lifts their cues upward while controlling reliability. We form a message $\hat{G}_{i+1} = \text{Conv}\downarrow(G_i) + \text{Resize}\downarrow(G_i)$. A channel attention block computes a spatial mask via channel pooling $M_{i+1} = \sigma\big(\text{AvgPool}(\hat{G}_{i+1}) + \text{MaxPool}(\hat{G}_{i+1})\big)$, and refine the message: $\hat{G}_{i+1} \leftarrow \hat{G}_{i+1} \odot M_{i+1}$. From $z_{i+1} = [G_{i+1}, M_{i+1}]$, we use three $1\times1$ projections $\psi$ that yields: (i) a nonnegative injection *strength* map $\alpha_{i+1} = \text{softplus}(\psi_\alpha(z_{i+1}))$, (ii) a bounded injection *confidence* map $\rho_{i+1} = \sigma(\psi_\rho(z_{i+1}))$, and (iii) feature map $R_{i+1} = \tanh(\psi_R(\hat{G}_{i+1}))$. Here, $\text{softplus}$ enforces nonnegativity and $\tanh$ limits residual energy for stability. The final update is $G_{i+1} \leftarrow G_{i+1} + (\rho_{i+1} \odot \alpha_{i+1}) \odot R_{i+1}$. This uncertainty-aware gating mechanism is, to our knowledge, novel technique in FR IQA; it is parameter-light, tolerates small misalignments, and exposes interpretable reliability maps.

**Spatial Cross-Gating Block (SCGB).** Inspired by (He et al., 2024), SCGB routes coarse context downward to suppress spurious high-resolution noise. For adjacent scales $i$ (fine) and $i+1$ (coarse), we form $g_{i\leftarrow i+1} = \sigma(\text{MLP}(G_{i+1}))$ and refine $G_i \leftarrow G_i + G_i \odot g_{i\leftarrow i+1}$. Together, upward CSRAM and downward SCGB provide a two-way highway that transfers only distortion evidence relevant to perceived quality.

**Reliability-Aware Scale-Wise Fusion.** BiRQA produces multi-scale feature tensors after SCGB, yet common fusion schemes (concatenations/MLPs or gating) hide cross-scale contributions and may be unstable across datasets. We introduce a light additive scale aggregation head that makes per-scale contributions explicit and learnable while keeping runtime overhead minimal. Post-SCGB tensors $G_i \in \mathbb{R}^{C \times H_i \times W_i}$ are fed into **Reliability-Aware Head** (**RAH**): each scale is pooled to $z_i = \text{GeM}(h_i(G_i)) \in \mathbb{R}^d$, where $h_i$ is a $1 \times 1$ conv to a shared width $d$, and GeM is generalized-mean pooling with learnable exponent $p$. Two tiny MLPs produce a contribution $c_i \in \mathbb{R}$ and a reliability logit $a_i \in \mathbb{R}$. Denoting $S$ as the number of scales, we form normalized gates $w_i = \text{softmax}(a_0, \ldots, a_{S-1})$ to obtain $\hat{y} = \sum_{i=0}^{S-1} w_i c_i$, a convex, interpretable aggregation that is stable across datasets and cheap to compute.

### 3.3. Anchored Adversarial Training

We propose an adversarial fine-tuning strategy, AAT, that leverages adversarial examples without directly penalizing quality labels. Our approach relies on the assumption that adversarial perturbations are either imperceptible or only slightly visible, which is the most common case in real-world adversarial scenarios. A key idea is to leverage the availability of clean examples (images with reliably predicted quality) as "anchors" for the training process.

**Problem Formulation.** Image-processing systems increasingly operate in untrusted environments where imperceptible perturbations can manipulate quality scores. We formalize an adversary that adds an $\ell_p$-bounded noise $\delta$ ($\|\delta\|_p \leq \epsilon$) to the distorted image $x_d$ of a pair $(x_r, x_d)$. When higher scores denote better quality the attacker seeks to maximize $f_\theta(x_r, x_d + \delta)$; if lower scores indicate higher quality, the attacker aims to decrease it. The same approach could do the opposite task, as demonstrated in (Antsiferova et al., 2024). This focus does not restrict the generality of our study, as the principles apply symmetrically. Formally, we define an adversarial attack as

$$\max_{\|\delta\|_p \leq \epsilon} f_\theta(x_r, \ x_d + \delta). \tag{1}$$

For a given model $f_\theta$; training data $D$ containing image pairs with associated quality label $y$; and a loss function $\mathcal{L}$ of the model, vanilla adversarial training is a min-max optimization problem((Chistyakova et al., 2024)):

$$\min_\theta \mathbb{E}_{(x_r, x_d, y) \sim D}\Big[ \max_{\|\delta\|_p \leq \epsilon} \mathcal{L}(f_\theta(x_r, x_d + \delta), y)\Big]. \tag{2}$$

The inner maximization generates strong adversarial examples $x_d + \delta$, while the outer minimization adjusts the model parameters $\theta$ to improve robustness.

In image classification, the ground-truth label is unaffected by adversarial perturbations. Quality scores, however, *do change* slightly with perceptual content, so directly re-using (2) creates a label-shift. Prior work tackles this by penalizing or rescaling the label (Chistyakova et al. 2024), but this brings two practical obstacles: (i) subjective studies follow diverse protocols, so matching MOS (Mean Opinion Score) scales must be repeated for every dataset; (ii) the metric that drives penalization can itself be vulnerable, enabling attack transfer.

## 4. Pointwise Error Bound via Anchored Ranking Loss

For a mini-batch of size $N$, let $y = (y_1, \ldots, y_N) \in \mathbb{R}^N$ be MOS labels (higher is better) and $\tilde{y} = (\tilde{y}_1, \ldots, \tilde{y}_N)$ be the corresponding model predictions. Let $\mathcal{S} \subseteq \{1, \ldots, N\}$ be the set of anchor indices. Anchor elements will not be attacked, and, thus, will have a reliable model predictions. Define the batch MOS range $R$ and maximum point-wise error $E := \|\tilde{y} - y\|_\infty = \max_{j \in [N]} |\tilde{y}_j - y_j|$. We also use $(t)_+ := \max\{0, t\}$.

For each non-anchor $j \notin \mathcal{S}$ choose nearest anchor elements

$$i^-(j) \in \arg\max_{i \in \mathcal{S}: y_i \leq y_j} y_i, \qquad i^+(j) \in \arg\min_{i \in \mathcal{S}: y_i \geq y_j} y_i,$$

and define one-sided ranking violations

$$v_j^+ := (\tilde{y}_j - \tilde{y}_{i^+(j)})_+, \qquad v_j^- := (\tilde{y}_{i^-(j)} - \tilde{y}_j)_+.$$

The per-sample loss and its batch max are

$$\ell_j^{\text{near}} := \frac{1}{R} \max\{v_j^+, v_j^-\}, L_{\text{anchor}}(y, \tilde{y}) := \max_{j \notin \mathcal{S}} \ell_j^{\text{near}}. \tag{3}$$

---

**Theorem 1: Pointwise Error Bound via Anchored Ranking Loss**

**Assumptions.**
1. **Anchor accuracy.** For all $i \in \mathcal{S}, |\tilde{y}_i - y_i| \leq \varepsilon$.

2. **Two-sided coverage.** For each $j \notin \mathcal{S}$, there exist anchors $i^-(j), i^+(j) \in \mathcal{S}$ such that

$$y_{i^-(j)} \leq y_j \leq y_{i^+(j)}, |y_j - y_{i^{\pm}(j)}| \leq \eta.$$

3. **Max-hinge control.** The anchored ranking loss satisfies $L_{\text{anchor}}(y, \tilde{y}) \leq \delta$.

Under these assumptions, the $E$ is bounded by

$$E \leq \varepsilon + \eta + R\delta. \tag{4}$$

---

**Proof sketch.** For any $j \notin \mathcal{S}$, (iii) gives $v_j^+ \leq R\delta$ and $v_j^- \leq R\delta$, hence

$$\tilde{y}_j \leq \tilde{y}_{i^+(j)} + R\delta, \qquad \tilde{y}_j \geq \tilde{y}_{i^-(j)} - R\delta.$$

Using (i) to replace $\tilde{y}_{i^{\pm}(j)}$ by $y_{i^{\pm}(j)} \pm \varepsilon$ and (ii) to relate $y_{i^{\pm}(j)}$ to $y_j \pm \eta$ yields $|\tilde{y}_j - y_j| \leq \varepsilon + \eta + R\delta$ for all non-anchors; anchors satisfy $|\tilde{y}_i - y_i| \leq \varepsilon$ by (i). Taking the maximum over $j$ gives the claim.

---

**Numerical example for MOS values in $[0, 10]$**

MOS labels are in $[0, 10]$, so $R = 10$. Select anchors approximately uniformly across MOS (e.g., by sorting the batch by MOS and taking MOS quantiles), so that every non-anchor has an anchor within $\eta = 0.25$ above and below. Suppose anchors are fitted to within $\varepsilon = 0.1$ MOS points, and during optimization we have $L_{\text{anchor}} \leq \delta = 0.01$. Then Theorem 1 gives

$$E \leq 0.1 + 0.25 + 10 \cdot 0.01 = 0.45.$$

---

**Notes on practicality of the assumptions.** Theorem 1 is intended to be *operational*: its conditions can be met by batching and optimization choices rather than dataset-specific calibration. Condition **A2** (two-sided coverage) is satisfied by sorting a mini-batch by MOS and selecting anchors uniformly across the MOS range (or via MOS quantiles), while ensuring the batch minimum and maximum MOS are included among anchors (to cover boundary

points). Condition **A1** (anchor accuracy) is enforced directly with a standard regression term on anchors (e.g., $\ell_1$ or $\ell_2$), and is empirically easy to monitor. Condition **A3** (max-hinge control) can be approximated in practice by using a top-$k$ surrogate to the max so that the largest violations are explicitly driven down. No dataset-specific statistics, MOS rescaling, or certified bounds are needed, so the conditions hold for any FR IQA corpus used in practice. Further details on the empirical satisfaction of these assumptions and on loss convergence are provided in Appendix C.2.

### 4.1. Implementation Details

For the vanilla BiRQA model we use Adam with $lr = 10^{-4}$ and batch size 32 and the following loss with $\alpha = 0.7$:

$$\mathcal{L}(y, \hat{y}) = \alpha \, MSE(y, \hat{y}) - (1 - \alpha) \, PLCC(y, \hat{y}). \tag{5}$$

During AAT, each mini-batch is constructed to have bounded label spread $R$ and dense anchor coverage. Within the batch, a subset of samples is kept clean to serve as anchors, while the remaining samples are adversarially perturbed. This makes the assumptions of Theorem 1 verifiable in practice: anchor predictions remain accurate (small $\varepsilon$), and the band construction ensures every non-anchor is bracketed by nearby anchors (small $\eta$).

Using Theorem 1 notation, let $J = j \notin \mathcal{S}$ and $\ell_{(1)} \geq \cdots \geq \ell_{(|J|)}$ be $\{\ell_j^{\text{near}} : j \notin \mathcal{S}\}$ sorted in descending order. For AAT we directly optimise a weighted mix of base and anchor loss functions:

$$\mathcal{L}_{\text{AAT}} = \tfrac{1}{2}\mathcal{L} + \tfrac{1}{2}\mathcal{L}_{\text{anchor}}; \mathcal{L}_{\text{anchor}} = \frac{1}{k} \sum_{r=1}^{k} \ell_{(r)}. \tag{6}$$

This loss directly targets the quantities appearing in Theorem 1: $\mathcal{L}$ drives anchor error $\varepsilon$, while the mined nearest-anchor hinge drives down the top-$k$ worst violations and approximates $L_{\text{anchor}}$. The complete set of parameters for BiRQA and AAT is listed in Appendix E.

**Mini-batch construction procedure.** In all AAT-BiRQA experiments we fix the number of anchors per mini-batch to $|\mathcal{S}| = 8$ with batch size $N = 16$. Let $MOS_r = max(MOS) - min(MOS)$ be the dataset MOS range; we set $R := \frac{MOS_r}{10} * \frac{|\mathcal{S}|}{N} = \frac{MOS_r}{20}$. For each batch, we first sample a contiguous MOS band of width $R$: we draw a "low" MOS value $y_{low}$ and form the band $[y_{low}, y_{low} + R]$. We then construct the anchors as follows: we always include the samples closest to $y_{low}$ and $y_{low} + R$ as anchors and then select the remaining $|\mathcal{S}| - 2$ anchors uniformly along the MOS axis inside the chosen band. By construction, the difference between any two consecutive anchors is at most $\eta$, which satisfies the coverage assumption in Theorem 1. Then we sample the remaining $N - |\mathcal{S}|$ non-anchor samples.

**Algorithm 1** Anchored Adversarial Fine-Tuning (AAT)

---

**Require:** Network $f_\theta$, training set $\mathcal{D}$, mini-batch size $N$, attack routine $A$, perturbation budget $\epsilon_{\mathrm{adv}}$, band width $R$, anchors per batch $|\mathcal{S}|$, top-$k$ parameter $k$

  **while** not converged **do**

    Sample mini-batch $\mathcal{B} = \{(x_r^m, x_d^m, y^m)\}_{m=1}^N$ from $\mathcal{D}$ as in Sec. 4.1

    Sort $\mathcal{B}$ by $y^m$ and select anchors $\mathcal{S}$ (quantiles + endpoints) {ensures coverage}

    **for** $m = 1, \ldots, N$ **do**

      **if** $m \in \mathcal{S}$ **then**

        $\hat{x}_d^m \leftarrow x_d^m$              {anchors remain clean}

      **else**

        $\hat{x}_d^m \leftarrow A(x_r^m, x_d^m; f_\theta, \epsilon_{\mathrm{adv}})$     {solve (1)}

      **end if**

      $\tilde{y}^m \leftarrow f_\theta(x_r^m, \hat{x}_d^m)$

    **end for**

    Compute $\mathcal{L}_{\mathrm{AAT}}$ via (6)

    $\theta \leftarrow \mathrm{ADAM}(\theta, \nabla_\theta \mathcal{L}_{\mathrm{AAT}})$

  **end while**

---

This construction ensures two-sided coverage assumption in Theorem 1: every non-anchor lies between two anchors whose MOS values are within $\eta \approx \frac{R}{|\mathcal{S}|-1}$ above/below, so the additive term $\eta$ in Theorem 1 is small by design. Moreover, keeping $R$ fixed across batches avoids large fluctuations in normalization and stabilizes the hinge scale, while anchor optimization focuses training on the top-k (we use $k = 4$) worst violations that matter for the pointwise guarantee.

## 5. Evaluation setup

*Table 1.* Full-Reference IQA datasets used in our experiments.

| Dataset | Resolution | Ref. Images | Dist. Images | Ratings |
|---|---|---|---|---|
| LIVE | $768 \times 512$ | 29 | 779 | 25k |
| CSIQ | $512 \times 512$ | 30 | 866 | 5k |
| TID2013 | $512 \times 384$ | 25 | 3,000 | 524k |
| KADID-10k | $512 \times 384$ | 81 | 10,125 | 30.4k |
| PIPAL | $288 \times 288$ | 250 | 23,200 | 1.13M |
| PieAPP | $256 \times 256$ | 200 | 20,280 | 1M+ |

**Datasets.** To compare our approach with current solutions, we provide various experiment results. We conducted intra- and cross-dataset evaluations, thorough robustness comparison, and an ablation study. As shown in Table 1, we conduct experiments on several public IQA datasets: LIVE (Sheikh et al. 2006), CSIQ (Larson & Chandler 2010), TID2013 (Ponomarenko et al. 2013), KADID-10k (Lin et al. 2019), PIPAL (Jinjin et al. 2020), and two-alternative forced choice (2AFC) dataset: BAPPS (Zhang et al. 2018). When available, we used official splits for train/val/test parts and the mean value for 10 runs on random splits in 6:2:2 proportion. These splits are based on reference images to prevent content leakage.

**Evaluation Metrics.** We evaluate performance using two widely accepted correlation metrics for datasets with MOS values: Pearson's Linear Correlation Coefficient (PLCC) and Spearman's Rank-Order Correlation Coefficient (SROCC). Both metrics are in the range [-1, 1], with a positive value meaning a positive correlation. A larger SROCC indicates a more accurate ranking ability of the model, while a larger PLCC indicates a more accurate fitting ability of the model. We also use a paired bootstrap test (1k resamples) to assess if differences in SROCC are statistically significant.

**Adversarial Robustness Comparison Methodology.** We compare the effectiveness of the proposed adversarial training method for the BiRQA (proposed) and LPIPS models to keep consistency with previous works. KADID-10k train and test parts were used for adversarial training and testing, utilizing the Projected Gradient Descent with 10 iterations (PGD-10, Madry et al. 2017) as attack method. The attack budget was $\epsilon = 8/255$. We compare adversarial training techniques by SROCC and Integral Robustness Score (IR-Score, Chistyakova et al. 2024) on clean and attacked data.

The IR-Score assesses the model's ability to withstand perturbations of varying strengths, as recommended in (Carlini et al., 2019). Adversarial examples were generated with perturbation magnitudes $\epsilon \in \mathcal{E} = \{2, 4, 8, 10\}/255$. Scores were normalized using min-max scaling and mapped to a unified domain via neural optimal transport to account for distributional differences.

## 6. Results

**Full-Reference Benchmarks.** Across standard FR-IQA benchmarks (LIVE, CSIQ, TID2013, PieAPP, PIPAL), BiRQA matches or surpasses prior SOTA in both PLCC and SROCC (e.g., LIVE 0.989/0.988, CSIQ 0.981/0.979; see Tab. 6 in Appendix due to limited space). On PieAPP, correlation is slightly lower, likely due to its broader distortion coverage. Per-distortion analysis shows SROCC $\approx$0.90-0.95 for most categories, with reduced effectiveness on some specific types such as radial geometric transforms.

To assess the generalization capabilities of the proposed model, we performed cross-dataset evaluations. The model was trained on large KADID-10k and PIPAL datasets and tested on LIVE, CSIQ, and TID2013 datasets. The results are presented in Table 2. The proposed base BiRQA performs comparably to the TOPIQ model and exceeds it in 9 out of 12 experiments, highlighting its robust generalization across diverse datasets. AAT-BiRQA performs slightly worse than vanilla BiRQA, but, nonetheless, keeps up with the SOTA performance. This shows that our adversarial training achieves robustness with negligible cost to normal-case performance – a key advantage over many defenses.

*Table 2.* Cross-dataset performance on benchmarks (PLCC and SROCC are reported in corresponding columns for each benchmark). The best values are bolded, and the second best are underlined.

| Method | Trained on **KADID-10k** | | | | | | Trained on **PIPAL** | | | | | |
| | LIVE | | CSIQ | | TID2013 | | LIVE | | CSIQ | | TID2013 | |
|---|---|---|---|---|---|---|---|---|---|---|---|---|
| WaDIQaM-FR (Bosse et al., 2017) | 0.940 | 0.947 | 0.901 | 0.909 | 0.834 | 0.831 | 0.895 | 0.899 | 0.834 | 0.822 | 0.786 | 0.739 |
| PieAPP (Prashnani et al., 2018) | 0.908 | 0.919 | 0.877 | 0.892 | 0.859 | 0.876 | — | — | — | — | — | — |
| LPIPS-VGG (Zhang et al., 2018) | 0.934 | 0.932 | 0.896 | 0.876 | 0.749 | 0.670 | 0.901 | 0.893 | 0.857 | 0.858 | 0.790 | 0.760 |
| DISTS (Ding et al., 2020) | 0.954 | 0.954 | 0.928 | 0.929 | 0.855 | 0.830 | 0.906 | 0.915 | 0.862 | 0.859 | 0.803 | 0.765 |
| AHIQ (Lao et al., 2022) | 0.952 | 0.970 | 0.955 | 0.951 | 0.889 | 0.885 | 0.903 | 0.920 | 0.861 | 0.865 | 0.804 | 0.763 |
| TOPIQ (Chen et al., 2024) | 0.957 | 0.974 | 0.963 | **0.969** | 0.916 | 0.915 | **0.913** | **0.939** | 0.908 | 0.908 | 0.846 | 0.816 |
| BiRQA (ours) | **0.967** | **0.977** | **0.966** | 0.967 | **0.925** | **0.921** | 0.911 | 0.933 | **0.913** | **0.912** | **0.855** | **0.824** |
| AAT-BiRQA (ours) | 0.961 | 0.968 | 0.960 | 0.959 | 0.917 | 0.916 | 0.907 | 0.921 | 0.909 | 0.909 | 0.847 | 0.817 |

**Adversarial Robustness Comparison.** We benchmark four variants of three full-reference IQA (LPIPS, TOPIQ and the proposed BiRQA) against common $\ell_\infty$ white-box attacks. Compared methods are: (1) **base**: no adversarial training; (2) **R-**: vanilla adversarial training; (3) **AT-**: adversarial training with label smoothing (Chistyakova et al. 2024); (4) **AAT-** (ours): Anchored Adversarial Training. All AT and AAT models were optimized with a PGD-10 attack of budget $\epsilon = 8/255$. At test time we evaluate four unseen attacks: FGSM (Goodfellow et al. 2014), C&W (Carlini & Wagner 2017), AutoAttack (AA, Croce & Hein 2020) and the perceptual FACPA attack (Shumitskaya 2023). Robustness is measured by SROCC and IR-Score on the KADID-10k dataset.

Table 3 presents the results, which shows that AAT achieves state-of-the-art robustness and outperforms other approaches by 0.02-0.06 SROCC points and by similar margins on IR-Score. AAT also provides the best SROCC on unperturbed test set compared to other defense methods. Even without defense, BiRQA possesses more robustness compared to both LPIPS and TOPIQ, surpassing them by $0.02 - 0.03$ in terms of IR-Score. Although AAT requires more time during adversarial fine-tuning phase, it provides the best overall results. More experiments can be found in the Appendix C.3-C.4, including experiments on 2AFC datasets under White-Box and Black-Box attacks.

**Theoretical bounds in practice.** Our adversarial training uses an anchored ranking loss that ties each perturbed sample to a small set of clean anchor images within the mini-batch. Theorem 1 establishes that as this loss approaches zero, the maximum prediction error on any adversarial example is provably bounded by a small constant. In practice, the empirical errors respect this bound, as shown in Fig. 5b in Appendix. Moreover, the objective drops below $10^{-2}$ within 500 iterations, indicating stable, fast optimization (see Fig. 5a in Appendix).

**Statistical significance** was tested on the PIPAL by a paired bootstrap of SROCC differences (1k resamples). As Table 8 in Appendix shows, BiRQA exceeds every previous FR IQA metric, gaining up to 0.57 SROCC over PSNR and a positive but very small ∼0.003 over the strongest baselines AHIQ and TOPIQ (statistically significant on PIPAL due to the large sample size, though the margin is small in absolute terms), while the robust variant AAT-BiRQA sacrifices only 0.007.

**Computational Complexity.** We assessed the computational efficiency of each IQA model by executing 100 forward passes on 100 random images from the PDAP-HDDS (Liu et al. 2018) dataset and averaging the resulting runtimes. Experiments were performed on an NVIDIA A100 GPU (80 GB). Measurements are end-to-end: every operation required by model is considered, including the feature calculation for BiRQA.

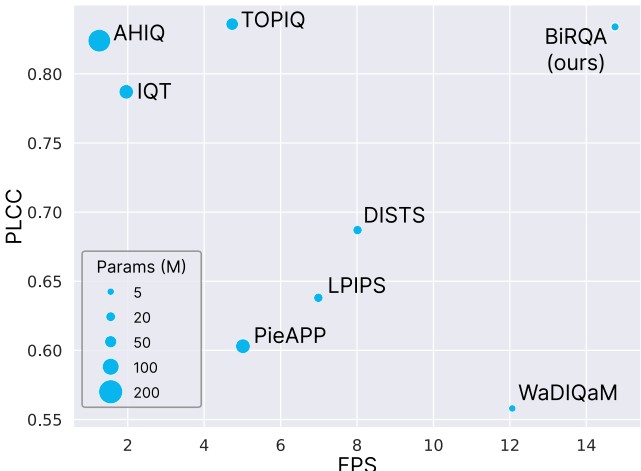

*Figure 3.* Computational efficiency (FPS) vs. Performance (PLCC) comparison on PDAP-HDDS dataset with image size of $1920 \times 1080$ pixels. Our model achieves comparable PLCC to SOTA method TOPIQ, while being ∼ 3 times faster and having a notably smaller number of parameters.

*Table 3.* Robustness of adversarially trained IQA models on KADID-10k. SROCC is evaluated at $\epsilon = 8/255$; IR-Score uses $\epsilon \in \{2, 4, 8, 10\}/255$. Bold numbers denote the best result for each model. All adversarial variants were trained only with PGD-10.

| Model | SROCC↑ | | | | | IR-Score↑ | | | | Train time |
| | Clean | FGSM | C&W | AutoAttack | FACPA | FGSM | C&W | AutoAttack | FACPA | (min) |
|---|---|---|---|---|---|---|---|---|---|---|
| base LPIPS | **0.893** | 0.542 | 0.260 | 0.239 | 0.496 | — | — | — | — | **46** |
| R-LPIPS | 0.858 | 0.570 | 0.327 | 0.266 | 0.515 | 0.541 | 0.403 | 0.385 | 0.507 | 123 |
| AT-LPIPS | 0.852 | 0.730 | 0.523 | 0.481 | 0.753 | 0.722 | 0.596 | 0.510 | 0.613 | 101 |
| AAT-LPIPS (ours) | 0.860 | **0.751** | **0.578** | **0.532** | **0.796** | **0.788** | **0.649** | **0.580** | **0.628** | 339 |
| base TOPIQ | **0.938** | 0.524 | 0.284 | 0.269 | 0.512 | — | — | — | — | **352** |
| R-TOPIQ | 0.879 | 0.533 | 0.305 | 0.314 | 0.509 | 0.572 | 0.431 | 0.456 | 0.520 | 437 |
| AT-TOPIQ | 0.892 | 0.839 | 0.513 | 0.552 | 0.760 | 0.763 | 0.615 | 0.550 | 0.611 | 514 |
| AAT-TOPIQ (ours) | 0.911 | **0.847** | **0.584** | **0.576** | **0.798** | **0.801** | **0.688** | **0.602** | **0.633** | 542 |
| base BiRQA | **0.954** | 0.568 | 0.295 | 0.350 | 0.503 | — | — | — | — | **105** |
| R-BiRQA | 0.902 | 0.571 | 0.291 | 0.357 | 0.502 | 0.563 | 0.427 | 0.414 | 0.525 | 218 |
| AT-BiRQA | 0.907 | 0.801 | 0.573 | 0.560 | 0.788 | 0.769 | 0.638 | 0.551 | 0.620 | 205 |
| AAT-BiRQA (ours) | 0.941 | **0.832** | **0.610** | **0.597** | **0.813** | **0.810** | **0.688** | **0.612** | **0.650** | 259 |

Figure 3 summarizes the trade-off between accuracy and efficiency. Our method achieves performance on par with TOPIQ while running substantially faster than all competing approaches. Figure 3 also reports the number of parameters for each model. BiRQA has 5.5M parameters, which is smaller than most of its counterparts, including LPIPS and TOPIQ.

**Ablation Study and Feature Selection.** We exhaustively trained 231 candidate models covering every combination of 1-, 2-, and 3-feature sets drawn from an 11-feature pool on KADID-10k. Each candidate was scored by a Pareto trade-off between SROCC and inference speed. We dropped the three weakest features and evaluated all 70 four-feature sets from the remaining eight. Validation SROCC flattened at four features; adding a fifth would increase inference cost without accuracy gains. The final combination (SSIM, Informational Map, Color Difference, and LBP) shows the best SROCC while remains relatively fast and captures complementary structure, information content, chromatic shifts, and fine texture cues. A model that uses only raw image pairs outperforms any single analytic feature map, but adding raw image input to the chosen four lowers SROCC, indicating the Color Difference map already embeds the raw signal. Complete ablations can be found in Appendix B.

Table 4 presents results on the KADID-10k dataset for the variation of the BiRQA model. Enabling CSRAM lifts correlations markedly, highlighting that cross-scale interchange with uncertainty-aware gating is crucial for capturing fine artifacts without losing global context. Adding SCGB then activates fully bidirectional information flow between scales and yields a further, consistent improvement, while the reliability-aware head consolidates these signals and sharpens calibration.

*Table 4.* Ablation study for BiRQA model. The CSRAM and SCGB modules were replaced with cross-attention layers and element-wise multiplication. The Reliability-Aware Head (RAH) module was replaced with pooling and MLP.

| CSRAM | SCGB | RAH | PLCC | SROCC |
|---|---|---|---|---|
| ✗ | ✗ | ✗ | 0.801 | 0.813 |
| ✓ | ✗ | ✗ | 0.907 | 0.911 |
| ✓ | ✓ | ✗ | 0.925 | 0.928 |
| ✓ | ✓ | ✓ | 0.938 | 0.942 |

## 7. Conclusion

In this work, we introduce BiRQA, a novel Full-Reference IQA metric that balances state-of-the-art performance while maintaining computational efficiency. It combines analytic feature maps with a lightweight neural network architecture, that incorporates a multi-scale pyramid framework, adaptive fusion and cross-scale attention mechanisms. This design captures multi-scale perceptual differences, achieving an inference speed of 15 FPS on $1920 \times 1080$ images, surpassing many counterparts. Extensive evaluations confirm that BiRQA matches or exceeds leading methods.

To address the critical challenge of adversarial robustness, we propose Anchored Adversarial Training (AAT) with an anchored-ranking loss and prove a mini-batch pointwise-error bound under mild assumptions. Empirically, the anchored loss drops quickly and remains low, and prediction errors stay within the theoretical bound. AAT delivers clear robustness gains over other defenses across four unseen attacks, with only a small drop on clean data. The trends persist when summarized by IR-Score across budgets. Limitations include weaker performance on some distortions (e.g., radial geometry). We hope these findings encourage further work into robust, efficient IQA models.

## Acknowledgements

The work of Aleksandr Gushchin and Anastasia Antsiferova were supported by a grant, provided by the Ministry of Economic Development of the Russian Federation (agreement dated June 20, 2025 No. 139-15-2025-011, identifier 000000C313925P4G0002). The research was carried out using the equipment of the Shared Research Facility "Shared Research Center of the Ivannikov Institute for System Programming of the Russian Academy of Sciences (SRC ISP RAS)" and the MSU-270 supercomputer of Lomonosov Moscow State University.

## Impact Statement

This paper presents work whose goal is to advance the field of Machine Learning. There are many potential societal consequences of our work, none which we feel must be specifically highlighted here.

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

# Appendix

## A. Proof of Theorem 1

**Notation.** Let $y = (y_1, \ldots, y_N) \in \mathbb{R}^N$ be MOS (higher is better) for a mini-batch of $N$ samples, and $\tilde{y} = (\tilde{y}_1, \ldots, \tilde{y}_N)$ be the corresponding model predictions. Let $\mathcal{S} \subseteq \{1, \ldots, N\}$ be the *anchor indices* and denote $M := |\mathcal{S}|$. Define the MOS range

$$R := \max_{i,j \in [N]} |y_i - y_j| \quad (> 0), \qquad E := \|\tilde{y} - y\|_\infty = \max_{j \in [N]} |\tilde{y}_j - y_j|.$$

**Two-sided anchor coverage.** Assume that for every $j \notin \mathcal{S}$ we can choose (possibly non-unique) anchors $i^-(j), i^+(j) \in \mathcal{S}$ such that

$$y_{i^-(j)} \le y_j \le y_{i^+(j)}, \qquad y_j - y_{i^-(j)} \le \eta, \qquad y_{i^+(j)} - y_j \le \eta, \tag{7}$$

for some $\eta \ge 0$. (In practice, $i^-(j), i^+(j)$ can be chosen as the nearest anchors in MOS below/above $y_j$.)

**Max-near anchored hinge loss.** For each non-anchor sample $j \notin \mathcal{S}$ define the *nearest-anchor violation magnitudes*

$$v_j^+ := (\tilde{y}_j - \tilde{y}_{i^+(j)})_+, \qquad v_j^- := (\tilde{y}_{i^-(j)} - \tilde{y}_j)_+,$$

and the (normalized) per-sample loss

$$\ell_j^{\text{near}}(y, \tilde{y}) := \frac{1}{R} \max\{v_j^+, v_j^-\}.$$

We then aggregate by the max over the batch

$$L_{\text{anchor}}(y, \tilde{y}) := \max_{j \notin \mathcal{S}} \ell_j^{\text{near}}(y, \tilde{y}).$$

(Anchors can be excluded from the max; their accuracy is controlled separately.)

**Theorem A.1** (Pointwise $\ell_\infty$ bound from max-near anchored hinge). *Assume:*

**(i) Anchor accuracy:** $|\tilde{y}_i - y_i| \le \varepsilon$ *for all* $i \in \mathcal{S}$.

**(ii) Two-sided coverage:** *the maps* $i^-(j), i^+(j)$ *exist and satisfy* (7) *for all* $j \notin \mathcal{S}$ *(with some* $\eta \ge 0$*).*

**(iii) Max-column control:** $L_{\text{anchor}}(y, \tilde{y}) \le \delta$.

*Then*

$$E \le \varepsilon + \eta + R\delta. \tag{8}$$

*Moreover, if the worst-error index* $j^\star \in \arg\max_j |\tilde{y}_j - y_j|$ *happens to be an anchor (* $j^\star \in \mathcal{S}$ *), then* $E \le \varepsilon$.

*Proof.* Fix any $j \notin \mathcal{S}$. From $L_{\text{anchor}} \le \delta$ we have

$$\max\{v_j^+, v_j^-\} \le R\delta, \quad \text{hence} \quad v_j^+ \le R\delta, \ v_j^- \le R\delta.$$

The inequality $v_j^+ = |\tilde{y}_j - \tilde{y}_{i^+(j)}| \le R\delta$ implies $\tilde{y}_j \le \tilde{y}_{i^+(j)} + R\delta$. By anchor accuracy, $\tilde{y}_{i^+(j)} \le y_{i^+(j)} + \varepsilon$. By coverage, $y_{i^+(j)} \le y_j + \eta$. Therefore

$$\tilde{y}_j \le (y_j + \eta) + \varepsilon + R\delta \quad \Rightarrow \quad \tilde{y}_j - y_j \le \eta + \varepsilon + R\delta.$$

Similarly, $v_j^- = |\tilde{y}_{i^-(j)} - \tilde{y}_j| \le R\delta$ implies $\tilde{y}_j \ge \tilde{y}_{i^-(j)} - R\delta$. By anchor accuracy, $\tilde{y}_{i^-(j)} \ge y_{i^-(j)} - \varepsilon$. By coverage, $y_{i^-(j)} \ge y_j - \eta$. Hence

$$\tilde{y}_j \ge (y_j - \eta) - \varepsilon - R\delta \quad \Rightarrow \quad y_j - \tilde{y}_j \le \eta + \varepsilon + R\delta.$$

Combining the two one-sided bounds yields $|\tilde{y}_j - y_j| \le \eta + \varepsilon + R\delta$ for all $j \notin \mathcal{S}$. For anchors $i \in \mathcal{S}$, anchor accuracy gives $|\tilde{y}_i - y_i| \le \varepsilon$. Taking the maximum over all $j \in [N]$ gives (8). If the maximizer lies in $\mathcal{S}$, then $E \le \varepsilon$. $\square$

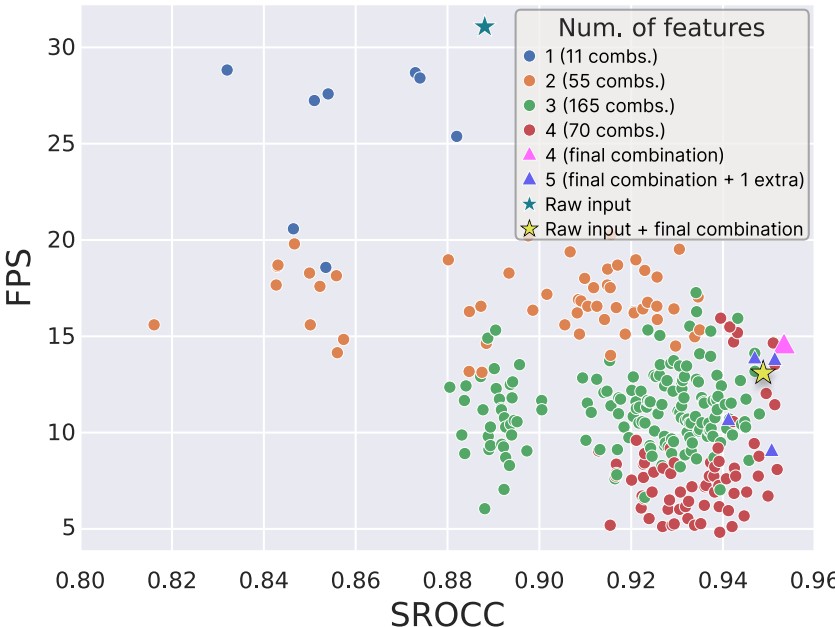

*Figure 4.* SROCC and inference FPS on KADID-10k for different feature sets. The chosen quartet lies on the Pareto frontier, offering the best accuracy. FPS was measured on images with $1920 \times 1080$ resolution.

## B. Choice of Features

We carefully designed a list of possible features for our model, including SSIM, Informational Map, Color Difference, Local Binary Pattern (LBP), Gabor Filters, Wavelet Transform, Entropy Map, Edge Map, Detail Loss Measure (DLM), Visual Information Fidelity (VIF) and Saliency Map. Short descriptions for these features are provided in Table 5.

We began with these 11 candidate analytic feature maps and exhaustively evaluated all 1-, 2-, and 3-feature combinations (11, 55, and 165 models, respectively) on KADID-10k (60/20/20 train/val/test split), measuring SROCC and inference speed (FPS). FPS was measured end-to-end (feature computation + the neural network) at $1920 \times 1080$ resolution on a single NVIDIA A100 80 GB GPU. Each model was trained from scratch under the same architecture and hyperparameters; $\approx 2$ GPU-hours per run, totaling $\approx 600$ GPU-hours. Based on joint accuracy–speed, the three weakest features (Saliency, Wavelet Transform, and Entropy Map) were removed. From the remaining eight features, we trained all $\binom{8}{4} = 70$ four-feature combinations. We did not explore all five-feature sets: their cost is prohibitive for real-time deployment and validation SROCC already saturates at four features (adding any additional feature to the best four-feature set yields no improvement). As a check, we added each of the remaining features to the best four-feature combination and tested them. The resulting SROCC values decreased upon adding a fifth feature, likely due to redundancy. In these experiments we varied only the input features to BiRQA, keeping the architecture fixed.

Figure 4 presents all 300+ evaluated feature combinations. The selected features SSIM, Informational Map (IM), Color Difference (CD), and LBP are complementary: SSIM captures structural deviations, IM reflects local information content, CD measures chromatic/luminance discrepancies in a perceptually motivated space, and LBP encodes fine-scale texture. A model using only raw image pairs outperforms any single analytic feature. However, concatenating raw pixels with the four selected features reduces SROCC. This likely stems from redundancy, because CD already encodes pixel level differences in an alternative color space, resulting in increased input dimensionality under a fixed capacity predictor, which can hurt generalization.

## C. Additional Analysis and Results

### C.1. More Results on FR Benchmarks

Table 6 presents a comprehensive comparison across widely used FR IQA benchmarks. The proposed BiRQA model achieves state-of-the-art or competitive results on most datasets, including LIVE, CSIQ, PieAPP, and the large-scale PIPAL

*Table 5.* Description of evaluated IQA features

| Feature | Description / Key Benefit |
| --- | --- |
| SSIM | Structural Similarity Index compares luminance, contrast and structural components between a reference and a test image, yielding a single score that tracks perceived quality with high correlation to the human visual system (HVS). |
| Informational Map | Generates a spatial weighting map based on local information content (gradient magnitude), giving more influence to perceptually important, detail-rich regions. |
| Color Difference | Computes perceptual color difference in YCbCr color space, making the metric sensitive to chromatic distortions that luminance-only measures may miss. |
| LBP | Local Binary Patterns encode micro-texture by thresholding each neighborhood against its centre pixel; the histogram is gray-scale and rotation robust, providing a compact descriptor of fine texture changes. |
| Gabor Filters | A bank of Gabor kernels isolates edge and texture energy in specific frequency-orientation bands, capturing blur, ringing and other anisotropic artifacts. |
| Wavelet Transform | Discrete wavelet decomposition splits the image into multi-resolution sub-bands; analyzing coefficients across scales localizes blur or compression artifacts while preserving both spatial and frequency information. |
| Entropy Map | Computes local Shannon entropy inside sliding windows; high-entropy areas correspond to regions with greater visual information, enabling quality scores that prioritize complex, information-dense regions. |
| Edge Map | Gradient-based edge extraction detects intensity discontinuities and object boundaries; comparing edge strength between reference and distorted images is effective at spotting blur or sharpening artifacts. |
| Detail Loss Measure (DLM) | Measures the dissimilarity of high-frequency gradients between image pairs, providing a direct quantification of lost fine-scale detail (e.g., due to denoising, compression or over-smoothing). |
| VIF | Visual Information Fidelity models images as Gaussian Scale Mixtures and computes the mutual information lost in the distorted version, grounding the metric in natural-scene statistics and information theory. |
| Saliency | Uses the saliency neural network model to predict likely gaze locations, helping the final metric align with where observers are most likely to look. |

dataset, for both PLCC and SROCC metrics. Notably, BiRQA reaches or surpasses a correlation of 0.98 on LIVE and CSIQ, and outperforms recent transformer-based methods on the more challenging PieAPP and PIPAL datasets while maintaining high efficiency.

AAT-BiRQA, which incorporates our proposed anchored adversarial training scheme, offers slightly lower correlations on clean data due to the regularization effect of adversarial robustness, but still maintains strong overall performance. This makes it preferable in safety-critical or attack-prone environments.

BiRQA shows slightly lower scores on TID2013, where it ranks just below the best-performing model (AHIQ). This can be attributed to the peculiar distortion types present in TID2013 such as chromatic aberration, mean shift, and severe radial distortions, that are underrepresented in modern training sets. Additionally, some distortions in TID2013 are known to interact poorly with analytic features (e.g., SSIM and LBP), which may limit BiRQA's ability to fully capture perceptual degradation in those cases. We hypothesize that deeper fine-tuning or explicit modeling of these artifact types may further improve performance on such legacy datasets.

Overall, BiRQA and its adversarially trained variant AAT-BiRQA demonstrate strong generalization across datasets and distortion types, validating the effectiveness of bidirectional multiscale fusion and anchor-based adversarial training.

*Table 6.* Quantitative comparison with related works on public FR benchmarks, including the traditional LIVE, CSIQ, TID2013 with MOS labels, and recent large-scale datasets PieAPP, PIPAL with 2AFC labels. The best and second results are bold and underlined, respectively, and "—" indicates the score is not available or not applicable.

| Method | LIVE | | CSIQ | | TID2013 | | PieAPP | | PIPAL | |
|---|---|---|---|---|---|---|---|---|---|---|
| | PLCC | SROCC | PLCC | SROCC | PLCC | SROCC | PLCC | SROCC | PLCC | SROCC |
| PSNR | 0.865 | 0.873 | 0.819 | 0.810 | 0.677 | 0.687 | 0.135 | 0.219 | 0.277 | 0.249 |
| SSIM (2004) | 0.937 | 0.948 | 0.852 | 0.865 | 0.777 | 0.727 | 0.245 | 0.316 | 0.391 | 0.361 |
| MS-SSIM (2003) | 0.940 | 0.951 | 0.889 | 0.906 | 0.830 | 0.786 | 0.051 | 0.321 | 0.163 | 0.369 |
| VIF (2006) | 0.960 | 0.964 | 0.913 | 0.911 | 0.771 | 0.677 | 0.250 | 0.212 | 0.479 | 0.397 |
| MAD (2010) | 0.968 | 0.967 | 0.950 | 0.947 | 0.827 | 0.781 | 0.231 | 0.304 | 0.580 | 0.543 |
| VSI (2014) | 0.948 | 0.952 | 0.928 | 0.942 | 0.900 | 0.897 | 0.364 | 0.361 | 0.517 | 0.458 |
| DeepQA (2017) | 0.982 | 0.981 | 0.965 | 0.961 | 0.947 | 0.939 | 0.172 | 0.252 | — | — |
| WaDIQaM (2017) | 0.980 | 0.970 | — | — | 0.946 | 0.940 | 0.439 | 0.352 | 0.548 | 0.553 |
| PieAPP (2018) | 0.986 | 0.977 | 0.975 | 0.973 | 0.946 | 0.945 | 0.842 | 0.831 | 0.597 | 0.607 |
| LPIPS-VGG (2018) | 0.978 | 0.972 | 0.970 | 0.967 | 0.944 | 0.936 | 0.654 | 0.641 | 0.633 | 0.595 |
| DISTS (2020) | 0.980 | 0.975 | 0.973 | 0.965 | 0.947 | 0.943 | 0.725 | 0.693 | 0.687 | 0.655 |
| JND-SalCAR (2020) | 0.987 | 0.984 | 0.977 | 0.976 | 0.956 | 0.949 | — | — | — | — |
| AHIQ (2022) | **0.989** | 0.984 | 0.978 | 0.975 | **0.968** | **0.962** | 0.840 | 0.838 | 0.823 | 0.813 |
| TOPIQ (2024) | 0.984 | 0.984 | 0.980 | 0.978 | 0.958 | 0.954 | 0.849 | 0.841 | 0.830 | 0.813 |
| BiRQA (ours) | **0.989** | **0.988** | **0.981** | **0.979** | 0.964 | 0.959 | **0.852** | **0.845** | **0.837** | **0.822** |
| AAT-BiRQA (ours) | 0.982 | 0.978 | 0.980 | 0.974 | 0.956 | 0.951 | 0.838 | 0.831 | 0.832 | 0.809 |

## C.2. Anchor-loss Convergence and Theoretical Bounds

Figure 5 (a) traces the anchored-ranking loss $\mathcal{L}_{anchor}$ over the first 1,000 optimization steps (mini-batches). Two curves are shown: the raw batch-wise loss (blue) and an exponential-moving-average with a window of 20 iterations (orange). By iteration ~650 the smoothed loss drops below $10^{-3}$ and remains there for the rest of training, with only small mini-batch jitter ($< 10\%$ relative amplitude). This meets the target $\delta = 10^{-3}$ used in Theorem 1, so the theoretical bound on pointwise prediction error is already guaranteed after less than two epochs. The plot confirms that anchored adversarial training converges quickly and stably, delivering the tight loss levels required for the robustness guarantee without extended hyper-parameter tuning.

Figure 5 (b) compares the theoretical bound of Theorem 1 with the observed maximum pointwise error during AAT fine-tuning on KADID-10k. The empirical curve never exceeds the bound and decays at the same exponential rate, giving concrete evidence that the bound is valid.

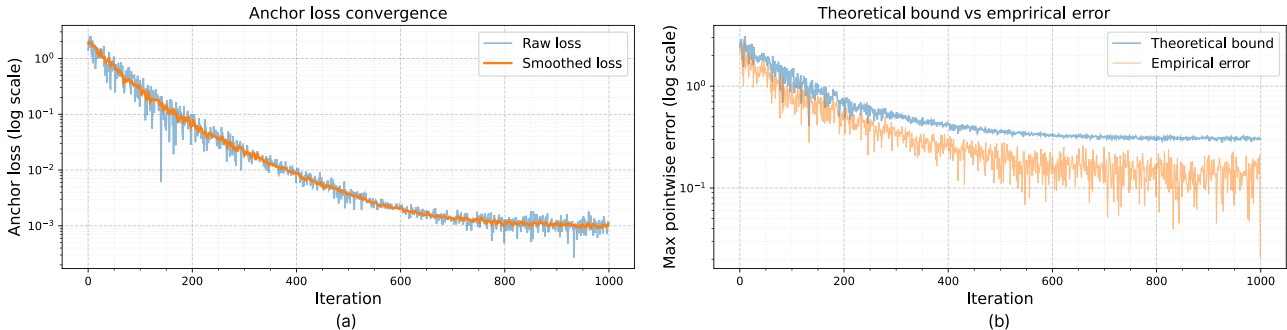

*Figure 5.* (a): Convergence of the Anchored Ranking Loss over 1,000 iterations. (b): Comparison of bounds, provided by Theorem 1 with empirical values of maximum pointwise errors.

## C.3. Robustness Under Different Attack Strengths

Figure 6 shows how Spearman rank-order correlation varies with FGSM attack strength on the KADID-10k test set. We evaluate BiRQA, LPIPS and TOPIQ together with their anchored adversarial training versions under five $\ell_\infty$ budgets $\epsilon = \{0, 2, 4, 8, 10\}/255$, where $\epsilon = 0$ corresponds to clean images. All models are trained on the KADID-10k training set. BiRQA consistently maintains higher correlation than LPIPS and TOPIQ as perturbation strength increases. Anchored adversarial training markedly reduces the performance drop for each metric, with the anchored BiRQA variant retaining the highest SROCC across all budgets.

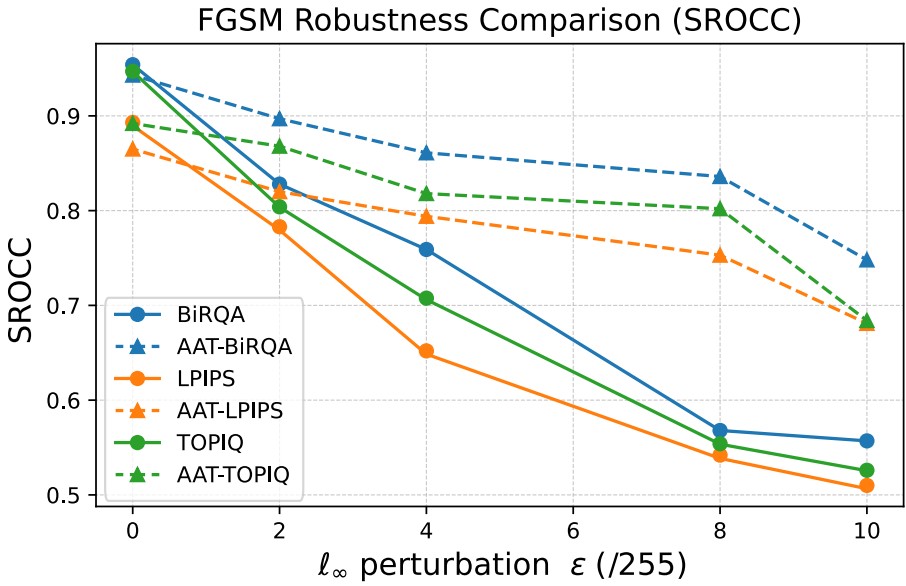

*Figure 6.* Robustness of FR IQA metrics to FGSM attack with different perturbation budgets evaluated on KADID-10k dataset. Solid lines show SROCC of vanilla models without adversarial training, dashed lines show SROCC of models trained with AAT.

## C.4. 2AFC Datasets Robustness Comparison

We evaluated four variations of three FR IQA models (LPIPS, TOPIQ and BiRQA) on Two Alternative Forced Choice (2AFC) dataset BAPPS. We compare **AAT** to (1) **base**: no adversarial training; (2) **R-**: vanilla adversarial training; (3) **AT-**: adversarial training with label smoothing. All AT and AAT models were optimized with a PGD-10 attack of budget $\epsilon = 8/255$. These models were attacked by seven unseen methods: FGSM, C&W, AutoAttack (AA), FACPA, the perceptual attack of Zhang et al., SquareAttack and Parsimonious, including 5 White Box and 2 Black Box methods. All attacks during testing were evaluated with a budget of $\epsilon = 8/255$. We report accuracy on clean and attacked versions of the BAPPS test set.

Table 7 presents the results, which shows that AAT achieves state-of-the-art robustness and outperforms other approaches by 0.02-0.1 accuracy points. AAT also provides the best accuracy on unperturbed test set compared to other defense methods. Even without defense, BiRQA possesses more robustness compared to LPIPS and TOPIQ, surpassing them by $0.03 - 0.06$ in terms of accuracy.

## C.5. Statistical Significance

We measure practical improvements by the difference in Spearman rank-order correlation (SROCC):

$$\Delta\rho_{i,j} = SROCC(y^{MOS}, y_i) - SROCC(y^{MOS}, y_j),$$

where $y^{\mathrm{MOS}}$ is the vector of mean-opinion scores, $i$ and $j$ index two IQA metrics, and $y_i$, $y_j$ are their predictions. For every pair of IQA metrics, we estimate $\Delta\rho$ and its 95% confidence interval (CI) via a paired non-parametric bootstrap: We used the PIPAL dataset with a size of $N = 23,200$ distorted images for this experiment. From the test set of $N$ image pairs we drew $R = 1,000$ bootstrap samples of size $N$ with replacement, keeping the MOS vector and the two prediction vectors

*Table 7.* Accuracy on 2AFC BAPPS test set of different adversarial training techniques, which were applied to LPIPS, TOPIQ and BiRQA models. The best results for each model are bolded. PGD-10 with $\epsilon = 8/255$ was used during training.

| Model | | | | White-Box Attacks | | | Black-Box Attacks | |
|---|---|---|---|---|---|---|---|---|
| | Clean | FGSM | C&W | AutoAttack | FACPA | Zhang et al. | SquareAttack | Parsimonious |
| base LPIPS | **0.742** | 0.260 | 0.102 | 0.135 | 0.415 | 0.301 | 0.511 | 0.513 |
| R-LPIPS | 0.728 | 0.487 | 0.306 | 0.298 | 0.471 | 0.375 | 0.567 | 0.533 |
| AT-LPIPS | 0.729 | 0.495 | 0.440 | 0.412 | 0.589 | 0.436 | 0.615 | 0.586 |
| AAT-LPIPS (ours) | 0.736 | **0.520** | **0.486** | **0.459** | **0.632** | **0.494** | **0.643** | **0.618** |
| base TOPIQ | **0.784** | 0.358 | 0.320 | 0.341 | 0.496 | 0.416 | 0.542 | 0.552 |
| R-TOPIQ | 0.762 | 0.420 | 0.391 | 0.350 | 0.523 | 0.459 | 0.589 | 0.581 |
| AT-TOPIQ | 0.768 | 0.493 | 0.467 | 0.479 | 0.584 | 0.510 | 0.634 | 0.614 |
| AAT-TOPIQ (ours) | 0.775 | **0.526** | **0.544** | **0.552** | **0.612** | **0.546** | **0.658** | **0.653** |
| base BiRQA | **0.794** | 0.405 | 0.350 | 0.376 | 0.521 | 0.462 | 0.571 | 0.574 |
| R-BiRQA | 0.771 | 0.491 | 0.382 | 0.417 | 0.545 | 0.526 | 0.614 | 0.587 |
| AT-BiRQA | 0.771 | 0.573 | 0.473 | 0.485 | 0.570 | 0.581 | 0.662 | 0.672 |
| AAT-BiRQA (ours) | 0.782 | **0.594** | **0.561** | **0.580** | **0.637** | **0.610** | **0.690** | **0.703** |

aligned. On each resample $b \in [1, ..., R]$ we computed $\Delta\rho^{(b)}$. The median of $\{\Delta\rho^{(b)}\}$ is reported as the effect size. The 2.5th and 97.5th percentiles form the two-sided 95% CI.

An improvement is considered significant when the lower CI bound is positive and $\Delta\rho \geq 0.01$, consistent with recent FR IQA benchmarks. Table 8 shows the results. The bootstrap makes no distributional assumptions, accounts for dependence between predictions, and remains valid even when error variance varies across the MOS range.

Table 8 shows that BiRQA outperforms every prior FR IQA metric on PIPAL. The gain is dramatic against classical metrics (e.g., +0.567 SROCC over PSNR) and even if small remains statistically significant against the strongest modern baselines (AHIQ/TOPIQ). The robustness-enhanced variant (AAT-BiRQA) sacrifices no more than 0.007 SROCC, confirming that anchored adversarial fine-tuning adds security almost for free.

*Table 8.* Pairwise ΔSROCC (±95% CI) on PIPAL (N = 23,000 distorted images) for 1,000 paired nonparametric bootstrap resamples (image pairs drawn with replacement; MOS and predictions kept aligned). Positive values favor the row metric; negative values favor the column metric. Only the lower triangle is shown; "—" indicates the symmetric counterpart.

| | PSNR | SSIM | MAD | WaDIQaM | LPIPS | DISTS | AHIQ | TOPIQ | BiRQA (ours) |
|---|---|---|---|---|---|---|---|---|---|
| PSNR | — | — | — | — | — | — | — | — | — |
| SSIM | 0.103 ±0.003 | — | — | — | — | — | — | — | — |
| MAD | 0.281 ±0.005 | 0.178 ±0.001 | — | — | — | — | — | — | — |
| WaDIQaM | 0.292 ±0.007 | 0.189 ±0.001 | 0.011 ±0.001 | — | — | — | — | — | — |
| LPIPS | 0.336 ±0.007 | 0.232 ±0.001 | 0.054 ±0.003 | 0.044 ±0.001 | — | — | — | — | — |
| DISTS | 0.399 ±0.006 | 0.295 ±0.001 | 0.117 ±0.006 | 0.107 ±0.004 | 0.063 ±0.002 | — | — | — | — |
| AHIQ | 0.564 ±0.017 | 0.460 ±0.014 | 0.282 ±0.014 | 0.272 ±0.012 | 0.228 ±0.009 | 0.165 ±0.004 | — | — | — |
| TOPIQ | 0.565 ±0.016 | 0.461 ±0.014 | 0.283 ±0.015 | 0.273 ±0.013 | 0.229 ±0.010 | 0.166 ±0.004 | 0.001 ±0.000 | — | — |
| BiRQA (ours) | 0.567 ±0.014 | 0.463 ±0.014 | 0.285 ±0.013 | 0.275 ±0.013 | 0.231 ±0.009 | 0.168 ±0.004 | 0.003 ±0.000 | 0.002 ±0.000 | — |
| AAT-BiRQA (ours) | 0.560 ±0.016 | 0.457 ±0.013 | 0.278 ±0.013 | 0.268 ±0.011 | 0.224 ±0.008 | 0.161 ±0.003 | −0.004 ±0.001 | −0.005 ±0.001 | −0.007 ±0.001 |

## D. Comparison with Purification Defenses

Table 9 compares the proposed AAT technique with adversarial purification methods. These methods do not modify the IQA model itself. Instead they are preprocess input images. We compare AAT-BiRQA against basic preprocessing techniques such as Random Flip, Random Rotate and the specialized DiffPure defense method ((Nie et al., 2022)). Results show that AAT-BiRQA outperforms all other purification methods, except DiffPure on SquareAttack. This likely reflects that SquareAttack has the least similar perturbations to PGD-10 used during adversarial training.

*Table 9.* Accuracy on 2AFC BAPPS test set for AAT-BiRQA compared to some adversarial purification methods. The best results are bolded. PGD-10 with $\epsilon = 8/255$ was used during training of AAT-BiRQA.

| Model | | White-Box Attacks | | | | | Black-Box Attacks | |
|---|---|---|---|---|---|---|---|---|
| | Clean | FGSM | C&W | AutoAttack | FACPA | Zhang et al. | SquareAttack | Parsimonious |
| base BiRQA | **0.794** | 0.405 | 0.350 | 0.376 | 0.521 | 0.462 | 0.571 | 0.574 |
| base BIRQA + Random Flip | 0.751 | 0.471 | 0.467 | 0.422 | 0.569 | 0.570 | 0.606 | 0.608 |
| base BIRQA + Random Rotate | 0.720 | 0.446 | 0.431 | 0.390 | 0.534 | 0.512 | 0.541 | 0.553 |
| base BIRQA + DiffPure | 0.741 | 0.522 | 0.498 | 0.452 | 0.619 | 0.603 | **0.712** | 0.698 |
| AAT-BiRQA (ours) | 0.782 | **0.594** | **0.561** | **0.580** | **0.637** | **0.610** | 0.690 | **0.703** |

## E. List of Parameters

The complete set of hyperparameters for both clean and adversarial training is provided in Table 10. For standard training, we use a regression-based objective that balances MSE and PLCC. In the adversarial setting, our anchored fine-tuning strategy integrates PGD-based attacks into the training loop and jointly optimizes clean and ranking losses. For the adversarial attacks and defenses, we have always used the default parameters except for $\epsilon$, whose value is explicitly stated.

*Table 10.* List of hyper-parameters and description of experimental set-up for BiRQA.

| Parameter | Value | Parameter | Value |
|---|---|---|---|
| *Vanilla (clean) training* | | *Anchored Adversarial Training (AAT)* | |
| Optimizer | Adam | Inner attacker | PGD-10 |
| Adam $(\beta_1, \beta_2)$ | 0.9/0.999 | PGD step size | 2/255 |
| Batch size | 32 | PGD norm type | $\ell_\infty$ |
| Epochs | 2500 | Perturbation budget $\epsilon$ | train: 8/255; test: $\{2, 4, 8, 10\}/255$ |
| Learning rate | $10^{-4}$ | Anchor spacing $\lambda$ | 0.5 |
| Loss function | $\mathcal{L}_{clean} = \alpha_1\text{MSE} - (1 - \alpha_1)\text{PLCC}$ $\alpha_1 = 0.7$ | AAT loss | $\alpha_2\mathcal{L}_{anchor} + (1 - \alpha_2)\mathcal{L}_{clean}$ $\alpha_2 = 0.5$ |

