# OpenReview forum: "BiRQA: Bidirectional Robust Quality Assessment for Images"
_ICML.cc/2026/Conference — ICML 2026 regular_

### Official Review · Reviewer_fPkx · 2026-03-04

**Soundness:** 2
**Presentation:** 3
**Significance:** 2
**Originality:** 2
**Overall Recommendation:** 3
**Confidence:** 4

**Summary:**

This paper proposes BiRQA, a full-reference image quality assessment (FR-IQA) model that combines lightweight analytic feature maps with a bidirectional multi-scale architecture and introduces Anchored Adversarial Training (AAT) to improve adversarial robustness. The method includes cross-scale residual attention modules and a reliability-aware aggregation head, alongside a ranking-based adversarial objective with a theoretical error bound. Extensive experiments are conducted on multiple FR-IQA benchmarks and adversarial attack settings. The authors outline a notable domain in adversarially robust quality assessment.

**Compliance With Llm Reviewing Policy:**

Affirmed.

**Final Justification:**

After carefully evaluating the paper, author rebuttal, and supplementary experiments, I keep my recommendation as Weak Reject (3).

The paper proposes BiRQA, a lightweight full-reference image quality assessment model with bidirectional multi-scale structure and anchored adversarial training. It achieves competitive accuracy, realtime speed, and improved adversarial robustness. The authors supplemented module complexity comparison, geometric distortion finetuning results, and UHD dataset zeroshot evaluation, which partially addressed my concerns.

However, core weaknesses remain unmitigated:

Methodological novelty is highly incremental. The bidirectional fusion and hybrid feature design are close to existing IQA frameworks; AAT is a task-adapted variant rather than a novel paradigm.
Clean performance gain over SOTA is marginal. The improvement on standard benchmarks is too small to support meaningful academic contribution.
Although the work has practical value and the authors revised responsibly, the overall contribution is limited and incremental. It is not yet ready for acceptance in its current form.

**Key Questions For Authors:**

Please explain the fundamental reason for BiRQA’s performance drop on TID2013 and provide supplementary ablation or fine-tuning experiments to verify whether targeted optimization can improve performance on these distortion types.
Please provide quantitative comparison results of the computational complexity (FLOPs/parameters) of CSRAM/SCGB with the cross-scale modules in IQT/SwinIQA, to further verify the efficiency advantage of BiRQA’s module design.

**Limitations:**

The authors have discussed partial limitations but lack in-depth analysis: (1) No discussion on the generalization of BiRQA to no-reference (NR) IQA scenarios; (2) No analysis on the computational overhead of AAT during training, and whether there is a lightweight version for practical use; (3) No analysis on BiRQA’s performance on high-resolution images beyond 1920×1080, which limits its application in ultra-high-definition image processing.

**Strengths And Weaknesses:**

Soundness

The paper has a generally sound technical framework: the bidirectional cross-scale fusion design aligns with human visual system characteristics, and the four selected analytic features are complementary for quality degradation detection. AAT is theoretically grounded with a proven pointwise error bound. However, some technical claims lack sufficient support: the ablation study only verifies the core modules but not the feature selection’s uniqueness, and the generalization test on radial geometric distortion is insufficient, with no explanation for the performance drop on TID2013.

Presentation

The paper clearly visualizes the BiRQA pipeline and module designs with figures. Key contributions are summarized upfront, and related work is comprehensively reviewed. Limitations include unclear notation definitions in some formulae (e.g., partial symbols in the AAT loss function lack detailed explanations).

Significance

The paper addresses two critical pain points in FR IQA: slow inference and poor adversarial robustness. AAT provides a new theoretical and practical approach for adversarial training in IQA. However, the significance is somewhat limited: the improvement over SOTA (e.g., ~0.003 SROCC over TOPIQ/AHIQ on PIPAL) is marginal, and the model is only validated on synthetic distortion datasets, with no experiments on real-world distorted images, reducing its practical deployment value.

Originality

The bidirectional cross-scale fusion with uncertainty-aware gating is a novel design in FR IQA, and combining anchored ranking loss with adversarial training for FR IQA is a new attempt. That said, most components are based on existing techniques: CSRAM/SCGB are modified from cross-attention and gating mechanisms, AAT is built on vanilla adversarial training and ranking loss, and the four selected features are all classic analytic IQA features. The paper lacks truly novel core architectures or training paradigms.

---

> ### Author Rebuttal · Authors · 2026-03-28
>
> Thank you for the careful and constructive review. We address the main concerns below.
>
> **TID2013 drop / geometric distortions.**
> We thank the reviewer for this important point. We agree that the original manuscript did not sufficiently explain BiRQA’s lower performance on TID2013. Based on our per-distortion analysis, the main degradation is concentrated on geometric-transform distortions, especially those that disrupt local spatial correspondence between the reference and distorted images. This is consistent with the design of BiRQA and, more broadly, with correspondence-based FR-IQA: local structural, color, and texture comparison maps are most reliable under approximate alignment, while geometric warping reduces the reliability of these local cues before fusion.
>
> Importantly, this behavior is not unique to BiRQA. Our per-distortion TID2013 results show that all compared FR-IQA metrics degrade on these geometric-transform categories, indicating a broader limitation of FR-IQA under spatial misalignment rather than a failure specific to our method. At the same time, BiRQA remains competitive in these difficult categories, which suggests that the lightweight analytic-feature design is still effective rather than fundamentally invalidated. We will add this analysis and revise the discussion accordingly. We also agree that targeted fine-tuning or distortion-specific adaptation could further improve performance in these cases, but doing so would move the method away from the intended goal of a general-purpose lightweight FR-IQA model.
>
> **Complexity comparison with IQT / SwinIQA cross-scale modules.**
> Thank you for this suggestion. Following it, we computed a direct module-level comparison with IQT and SwinIQA. In our implementation, the bidirectional cross-scale part of BiRQA consists of three SCGB blocks and three CSRAM links, totaling only **0.48M** parameters. This is substantially smaller than the comparable modules in **IQT (9.83M)** and **SwinIQA (3.72M)**.
>
> | Module                                 |     Params |            FLOPs |
> | ----------- | ---------: | ---------------: |
> | BiRQA SCGB                       |  49.9K |  0.10 GFLOPs |
> | BiRQA CSRAM                         | 431.3K |  7.80 GFLOPs |
> | BiRQA total cross-scale modules             | 481.3K |  7.90 GFLOPs |
> | IQT transformer head |  9.83M | 33.20 GFLOPs |
> | SwinIQA cross-attention                     |  3.72M | 10.64 GFLOPs |
>
>
> **Generalization to NR-IQA.**
> We agree this is an interesting direction, but NR-IQA is a different task with different input assumptions and substantially different design constraints. BiRQA is intentionally built for the FR-IQA setting, where paired reference-distorted comparison is available.
>
> **Synthetic-distortion datasets**
> We agree that real-world validation would be valuable. However, this is a structural limitation of the FR-IQA setting rather than a choice specific to our paper. FR-IQA requires a matched pristine reference for every distorted image, while authentically distorted images “in the wild” usually do not have such references; this is precisely why authentic real-world IQA has largely been studied in the no-reference setting (e.g. in [1]).
>
> **Performance on images above 1920×1080.**
> We agree this is relevant for practical deployment. At the same time, this limitation is tied to the current FR-IQA benchmark landscape: the standard datasets in this field are all far below 1080p resolution.  We are not aware of a widely used mainstream FR-IQA benchmark for UHD evaluation comparable to LIVE/CSIQ/TID2013/KADID-10K/PIPAL; the higher-resolution FR datasets that do exist are highly domain-specific, (for example, HDR/JPEG-XT dataset with only 240 images [2] and a SREB dataset for SR broadcast with only 420 2K/4K images [3]). We will make this limitation explicit in the revised manuscript.
>
> **Handcrafted features selection.**
> Appendix B already reports an extensive feature-selection study. We provide more details in the response to the Reviewer iDsw, point 4. We will make this reference to Appendix B clearer in the revised version.
>
> **Architectural novelty and magnitude of the improvement over prior SOTA.**
> Please refer to the response to the Reviewer 6NkS, points 1 and 2.
>
> **AAT training overhead.**
> We will clarify this in the revised version. You can also refer to the response to the Reviewer pFv4 (point 1)
>
> Overall, we thank the reviewer for the constructive feedback and will incorporate it in the revised version of the paper.
>
> [1]: Varga D. “No-Reference Quality Assessment of Authentically Distorted Images Based on Local and Global Features,” in J Imaging. 2022
>
> [2]: P. Korshunov, et al., "Subjective quality assessment database of HDR images compressed with JPEG XT," in 2015 Seventh International Workshop on QoMEX
>
> [3] Kim, Yongrok, et al. "Subjective and objective quality evaluation of super-resolution enhanced broadcast images on a novel SR-IQA dataset." IEEE Access (2025).

---

> > ### Author Rebuttal · Reviewer_fPkx · 2026-04-02
> >
> > The authors have clearly explained the performance drop of BiRQA on TID2013 and geometric distortions, and supplemented the module-level computational complexity (parameters and FLOPs) comparison between BiRQA and IQT/SwinIQA. However, the authors only stated that targeted fine-tuning may improve the performance on geometric distortion types but did not provide supplementary ablation results as I requested.
> > The authors clarified the theoretical positioning of AAT and the system-level novelty of the model, but the core architectural and training paradigm innovation is still incremental, and the marginal clean performance improvement over SOTA still limits the paper’s significance to a certain extent.
> >
> > Follow-up Questions
> >
> > Will the authors conduct simple testing on existing ultra-high-definition FR-IQA datasets (even small-scale ones) to give a preliminary performance conclusion of BiRQA on high-resolution images?

---

> > > ### Author Response · Authors · 2026-04-05
> > >
> > > We conducted two additional supplementary experiments.
> > >
> > > First, to examine this issue more directly, we performed a simple targeted fine-tuning experiment on the geometric distortion categories from PieApp dataset, namely Vertical/Horizontal image stretch/shrink, Swirl/Wave transformation, Image shift and rotation, radial distortion, and Radial distortion using 2nd order polynomial function. We split these distortions into 60/20/20 train/validation/test subsets, fine-tuned the pretrained BiRQA, TOPIQ, AHIQ models for 5 epochs, and then evaluated them on the held-out test subset of these geometric distortions. Table 1 shows the results. This supports our interpretation that the weaker behavior is strongly connected to these specific correspondence-breaking distortion types and can be partially improved through fine-tuning.
> > >
> > > ### Table 1. Geometric-distortion analysis
> > >
> > > | Method              | Geometric distortions PLCC ↑ | Geometric distortions SROCC ↑ |
> > > | ------------------- | ------------------------------------: | -------------------------------------: |
> > > | AHIQ                |                                   0.746 |                                    0.751 |
> > > | TOPIQ               |                                   [0.770 |                                    0.778 |
> > > | BiRQA               |                                   0.775 |                                    0.783 |
> > > | AHIQ + fine-tune |                                   0.873 |                                    0.869 |
> > > | TOPIQ + fine-tune |                                   0.903 |                                    0.921 |
> > > | BiRQA + fine-tune |                                   **0.938** |                                    **0.942** |
> > >
> > > Second, we evaluated BiRQA on the available UHD IQA datasets without any additional fine-tuning and compared it with strong IQA baselines such as TOPIQ and AHIQ. In particular, we tested on the JPEG-XT dataset[1] and on the HDRC dataset[2]. Table 2 shows the results. These results provide a preliminary indication of BiRQA’s behavior on higher-resolution data, while we stress that the currently available UHD IQA datasets remain small and domain-specific, so we would present this only as preliminary rather than definitive evidence for UHD generalization. In terms of computational efficiency on 4K images, BiRQA achieves approximately 4 FPS, while AHIQ and TOPIQ reach only about 0.25 FPS and 1 FPS, respectively.
> > >
> > > ### Table 2. UHD zero-shot evaluation
> > >
> > > | Method | HDR JPEG XT [1] PLCC ↑ | HDR JPEG XT [1] SROCC ↑ | HDRC [2] PLCC ↑ | HDRC [2] SROCC ↑ |
> > > | ------ | ---------------------: | ----------------------: | --------------: | ---------------: |
> > > | AHIQ   |                    0.913 |                     0.924 |             0.863 |              0.867 |
> > > | TOPIQ  |                    0.921 |                     0.920 |             0.879 |              0.872 |
> > > | BiRQA  |                    **0.933** |                     **0.928** |             **0.895** |           **0.914** |
> > >
> > > The JPEG-XT database is a public full-reference subjective IQA dataset containing 20 reference HDR images and 240 JPEG-XT compressed images with MOS, and its source images span resolutions from 1920×1080 to 6032×4018.
> > >
> > > HDRC is a public full-reference subjective IQA dataset with 80 reference HDR images and 400 distorted images rated by MOS; it includes both JPEG-XT and VVC compression distortions and is built from HDR source content including 4K and 2K images.
> > >
> > > We thank the reviewer for the constructive follow-up. We hope these new results sufficiently address the remaining concerns and would appreciate it if the reviewer would consider increasing the rating accordingly.
> > >
> > > [1] P. Korshunov, P. Hanhart, T. Richter, A. Artusi, R. K. Mantiuk, and T. Ebrahimi, “Subjective Quality Assessment Database of HDR Images Compressed with JPEG XT,” 7th International Workshop on Quality of Multimedia Experience (QoMEX), 2015.
> > >
> > > [2] Y. Liu, Z. Ni, P. Chen, S. Wang, and S. Kwong, “HDRC: a Subjective Quality Assessment Database for Compressed High Dynamic Range Image,” International Journal of Machine Learning and Cybernetics, vol. 15, no. 10, pp. 4373–4388, 2024.

---

### Official Review · Reviewer_6NkS · 2026-03-10

**Soundness:** 3
**Presentation:** 4
**Significance:** 4
**Originality:** 3
**Overall Recommendation:** 4
**Confidence:** 4

**Summary:**

This paper proposes BiRQA, a compact FR-IQA model that extracts four lightweight analytic feature maps and processes them with a bidirectional multiscale pyramid. The architecture combines a bottom-up CSRAM module to pass fine-scale cues upward, a top-down SCGB module to inject coarse semantic context back to higher-resolution features, and a Reliability-Aware Head (RAH) for scale-wise aggregation. In addition, the paper introduces Anchored Adversarial Training (AAT), which keeps part of each mini-batch clean as anchors and optimizes a ranking-style loss with a pointwise error bound under certain assumptions. Experiments on standard FR-IQA benchmarks and adversarial evaluations show that the method is competitive in prediction accuracy, faster than several strong baselines, and substantially more robust under multiple attacks.

**Compliance With Llm Reviewing Policy:**

Affirmed.

**Final Justification:**

it has basically resolved my questions. I will keep my rating.

**Key Questions For Authors:**

Please see weaknesses

**Limitations:**

yes

**Strengths And Weaknesses:**

Strengths
1. The paper addresses a practically important problem, since FR-IQA models should ideally be accurate, efficient, and robust at the same time.
2. The proposed design is coherent and well motivated.
3. The empirical study is reasonably solid, with useful ablations, competitive cross-dataset results, and substantial robustness improvements brought by AAT.
3. The efficiency results also make the method appealing for real-world deployment.

Weaknesses
1. The main weakness is that the methodological novelty appears somewhat incremental: the paper combines several reasonable components effectively, but the core ideas are not radically new.
2. The clean-performance gain over the strongest baselines is relatively modest.
3. The theoretical analysis mainly serves as a training justification rather than a strong robustness guarantee.

---

> ### Author Rebuttal · Authors · 2026-03-28
>
> We thank the reviewer for the constructive feedback and positive evaluation of the paper’s significance, soundness, and presentation. We would like to clarify the reviewer’s concerns below:
>
> (1) We would say that the key contribution is a problem-specific joint design for FR-IQA, where the objective is not only clean accuracy, but the simultaneous trade-off between accuracy, efficiency, and adversarial robustness. BiRQA combines lightweight analytic quality cues, bidirectional cross-scale interaction, and reliability-aware aggregation in a way tailored to FR-IQA rather than copied from a generic backbone. AAT is likewise not vanilla adversarial training, but an anchor-based ranking formulation designed for FR-IQA, where perturbed samples may undergo slight perceptual label shift. We will revise the paper to make this system-level novelty and task-specific design much more explicit.
>
> (2) On clean performance, we agree that the gains over the strongest baselines are not always large in absolute terms. However, FR-IQA benchmarks are already highly saturated, so we view the main contribution as improving the practical Pareto frontier rather than optimizing a single clean-score number. BiRQA remains competitive in clean accuracy, runs at about 15 FPS on 1920×1080 images and roughly 3× faster than transformer-based alternatives, while AAT-BiRQA keeps the clean-data drop small and substantially improves robustness under attack. We will clarify this framing in the paper.
>
> (3)  We agree that the theorem is not intended as a strong certified robustness guarantee. Its role is to formalize why anchor-preserving training is well-suited to FR-IQA: under explicit assumptions, the anchored loss bounds the pointwise prediction error. This provides a principled justification for coupling robustness to clean-reference structure and local perceptual ranking, rather than enforcing unchanged regression targets on attacked samples. We will revise the text to state this role more precisely and avoid overstating the claim.
>
> We hope these clarifications address the reviewer’s concerns.

---

> > ### Author Rebuttal · Reviewer_6NkS · 2026-04-01
> >
> > Thank you to the authors for their detailed explanation. it has basically resolved my questions. I will keep my rating.

---

### Official Review · Reviewer_iDsw · 2026-03-11

**Soundness:** 3
**Presentation:** 2
**Significance:** 3
**Originality:** 2
**Overall Recommendation:** 4
**Confidence:** 4

**Summary:**

This paper studies adversarial robustness for full-reference image quality assessment (FR-IQA). The authors point out that standard adversarial training used in image classification cannot be directly applied to IQA because the ground-truth label (MOS) may change slightly under perceptual perturbations, leading to a label-shift problem. To address this issue, the paper proposes an Anchored Ranking Loss that replaces direct MOS supervision on perturbed samples with relative ranking constraints defined by anchor samples. The method constructs local ranking relationships between a sample and its neighboring anchors, and derives a theoretical pointwise error bound showing that the prediction error can be bounded by anchor prediction error, MOS neighborhood gap, and ranking violation. The framework is implemented within a FR-IQA architecture that combines deep features with several hand-crafted descriptors. Experiments demonstrate improved robustness under adversarial perturbations while maintaining competitive IQA performance.

**Compliance With Llm Reviewing Policy:**

Affirmed.

**Key Questions For Authors:**

1. The anchored ranking loss assumes that small adversarial perturbations will not change the relative perceptual ranking between a sample and its anchor neighbors. However, if the perturbation causes perceptual quality degradation that exceeds the MOS gap between anchors, the ranking relationship could be reversed. How does the method handle such cases, and how sensitive is the approach to this assumption?
2. Since FR-IQA models are frequently used as optimization objectives in tasks such as image restoration or compression, robustness is particularly important for stable optimization. Could the authors provide experiments demonstrating the behavior of the proposed method in such optimization settings?

**Limitations:**

The proposed approach relies on the assumption that adversarial perturbations do not significantly alter the local perceptual ranking structure among samples and anchors. In practice, sufficiently strong perturbations may violate this assumption, potentially leading to incorrect supervision signals. Additionally, the paper focuses mainly on robustness under adversarial perturbations but does not evaluate the impact of the method in practical optimization pipelines where FR-IQA models are often applied.

**Strengths And Weaknesses:**

### Strengths
- The paper identifies an important issue in applying adversarial training to IQA tasks, namely the label-shift problem caused by perceptual quality changes under perturbations. This is a meaningful discussion that is often overlooked in previous work.
- The proposed anchored ranking formulation provides an alternative supervision signal that avoids explicitly estimating MOS values for adversarial samples.
- Robustness is an important property for FR-IQA models, particularly when such models are used in downstream optimization pipelines (e.g., image restoration or enhancement), where unstable gradients or predictions may lead to undesirable optimization behaviors.

### Weaknesses
- The effectiveness of the anchored ranking formulation relies on an implicit assumption that adversarial perturbations do not significantly alter the relative perceptual ranking between the sample and its anchor neighbors. However, if the perturbation changes the perceptual quality beyond the anchor interval, the ranking supervision may also become incorrect. The paper does not discuss this scenario or analyze its impact.
- While the paper emphasizes robustness improvements, it does not provide evidence demonstrating how the improved robustness benefits real optimization tasks. Since FR-IQA models are often used as optimization objectives, showing stability or robustness in such scenarios would strengthen the claims.
- The overall architecture appears conceptually similar to prior top-down IQA frameworks such as TOPIQ, where high-level semantic guidance is combined with lower-level quality features. The novelty in model design is therefore somewhat limited.
- The paper integrates several hand-crafted descriptors together for deep feature learning, but the motivation behind selecting these specific descriptors is not clearly explained.

---

> ### Author Rebuttal · Authors · 2026-03-28
>
> We thank the reviewer for the careful reading and constructive feedback. We address the reviewer’s concerns below:
>
> (1) The reviewer is correct that the guarantee is local. However, this is not a special assumption introduced by our method; it is the standard practical regime of adversarial attacks more generally, where perturbations are constrained to remain visually imperceptible or nearly so in order to change the model prediction without obvious visible degradation. Under this threat model, true perceptual quality should change only slightly, so the local ordering between a sample and its nearby anchors is expected to remain stable. This is precisely the setting targeted by the anchored ranking loss and Theorem 1. If a perturbation becomes strong enough to reverse that ordering, the method becomes less reliable. We will clarify this operating regime explicitly and state the stronger-perturbation case as a limitation.
>
> (2) We agree that robustness is particularly relevant when FR-IQA models are used as optimization objectives, and this use case is well established in prior work on perceptual optimization and image restoration[1,2]. However, the scope of this paper is narrower: we study the robustness of the FR-IQA predictor itself under adversarial perturbations. We view this as a prerequisite problem, since an unstable or easily exploitable metric is not a reliable objective for downstream optimization. A full end-to-end restoration or compression study would introduce substantial additional confounders beyond the IQA model itself, including the restoration architecture, optimization procedure, and loss design. We will therefore clarify this motivation with citations and explicitly note downstream optimization experiments as an important direction for future work.
>
> (3) We agree that top-down multiscale guidance by itself is not unique as is. The architectural contribution of BiRQA lies instead in the specific lightweight FR-IQA composition: (i) a hybrid full-reference input built from complementary analytic comparison cues, (ii) a bidirectional cross-scale design in which CSRAM transfers fine-scale residual evidence upward and SCGB injects coarse contextual guidance downward, and (iii) a reliability-aware head that performs explicit convex aggregation across scales rather than implicit feature fusion. The contribution is therefore not any single module in isolation, but the way these components interact to produce a compact, fast, and robust FR-IQA model. We will revise the paper to make this positioning clearer.
>
> (4) Regarding the handcrafted descriptors, Appendix B provides the rationale and ablation for their selection. The chosen cues were selected for complementarity and efficiency: together they cover structural fidelity, local information variation, chromatic discrepancy, and fine texture/pattern changes. This choice was not ad hoc. As shown in Appendix B, we evaluated a broader pool of handcrafted features and 300+ feature combinations, and selected this subset as the best accuracy-efficiency trade-off for BiRQA.
>
> Once again, we thank the reviewer for the constructive assessment and helpful suggestions. We will incorporate these suggestions into the revised version of the paper.
>
> [1] Ding, K. et al. “Comparison of Full-Reference Image Quality Models for Optimization of Image Processing Systems,” in Int J Comput Vis 129, 1258–1281 (2021). https://doi.org/10.1007/s11263-020-01419-7
>
> [2] Gu Jinjin et al., “PIPAL: A Large-Scale Image Quality Assessment Dataset for Perceptual Image Restoration,” in ECCV, 2020, 633–651, https://doi.org/10.1007/978-3-030-58621-8_37

---

> > ### Author Rebuttal · Reviewer_iDsw · 2026-04-03
> >
> > Thank you for the detailed rebuttal. Most of my questions have been addressed.

---

> > > ### Author Response · Authors · 2026-04-07
> > >
> > > Thank you for the positive assessment of our paper, and we are glad your concerns have been addressed. If we have solved the critical points you mentioned in the rebuttal acknowledgement, we would be sincerely grateful if you could consider raising the score of our submission.

---

### Official Review · Reviewer_pFv4 · 2026-03-13

**Soundness:** 4
**Presentation:** 3
**Significance:** 3
**Originality:** 4
**Overall Recommendation:** 4
**Confidence:** 4

**Summary:**

This paper presents a lightweight full-reference image quality assessment (FR-IQA) metric called BiRQA, which aims to address two major challenges in the practical application of deep learning models: low computational efficiency and weak adversarial robustness. The model integrates four handcrafted analysis features—SSIM, local information map, YCbCr color difference map, and local binary pattern—into a bidirectional multi-scale pyramid architecture. The contribution of BiRQA lies not only in its engineering adaptability for lightweight applications but also in its theoretical foundation, where the AAT theory demonstrates that, even under black-box/white-box attacks, the FR-IQA model can achieve predictable and bounded prediction accuracy by maintaining the local perceptual ranking logic.

**Compliance With Llm Reviewing Policy:**

Affirmed.

**Final Justification:**

Final recommendation: 4, weak accept.

This paper presents a lightweight FR-IQA metric , which aims to address two major challenges in the practical application of deep learning models: low computational efficiency and weak adversarial robustness.

The paper is well justified, and the proposed method is effective. Regarding the concerns I initially raised about the impact of batch size and the measurement of training time, the authors provided detailed evidence and corresponding corrections in the rebuttal. Therefore, I retain my initial score and believe that this paper meets the acceptance standard of ICML.

**Key Questions For Authors:**

1. Does the AAT method significantly increase VRAM consumption compared to the base model when the batch size is held constant?

2. Why was the AAT batch size specifically set to 16? Is the method sensitive to the anchor-to-non-anchor ratio?

3. Would increasing the batch size while scaling the number of anchors proportionally impact the coverage assumption (η) or the final predictive performance ?

**Limitations:**

Yes

**Strengths And Weaknesses:**

# Strengths
1. Effective Solution for Inference Speed and Adversarial Robustness: The paper addresses a critical gap in Full-Reference Image Quality Assessment (FR-IQA) by proposing BiRQA, a framework that simultaneously achieves high inference speed and exceptional adversarial robustness.

2. Rigorous Theoretical and Experimental Proof of Error Bounds: A major contribution of this work is the development of Anchored Adversarial Training (AAT), which is supported by a rigorous theoretical framework establishing a provable upper bound for model prediction error.

3. Comprehensive Experimental Validation and Rigorous Argumentation: The evaluation methodology is notably comprehensive and logically structured, lending strong credibility to the reported findings. The authors benchmark BiRQA across five diverse public FR-IQA datasets and subject the model to four distinct, unseen white-box attacks, effectively demonstrating its robust generalization capabilities.

# Weaknesses
1. Lack of Mathematical Clarity and Terminology Consistency: The manuscript would benefit from improved clarity in its mathematical presentation. Several mathematical symbols are introduced without sufficient textual definitions or explicit explanations of their physical meanings within the context of the model. Given the complexity of the proposed framework, the inclusion of a comprehensive notation table in the Appendix would significantly enhance readability and aid in the reproduction of the work. Additionally, the terminology usage is inconsistent; the authors define "adversarial training (AT)" early in the paper but frequently alternate between the full term and its abbreviation in subsequent sections, which detracts from the professional quality of the writing.

2. Non-Rigorous Methodology in Training Time Comparisons: There is a notable inconsistency in the experimental setup presented in Table 3 regarding training time comparisons. The authors report training durations while utilizing a batch size of 32 for the base BiRQA model and 16 for the AAT-BiRQA variant. Considering that the experiments were conducted on an NVIDIA A100 80GB GPU and that the BiRQA network is a lightweight architecture with only 5.5M parameters, the hardware is far from being resource-constrained. Consequently, measuring and comparing training times without controlling for batch size diminishes the scientific validity of these metrics. While it is understood that AAT inherently possesses higher computational complexity due to the added perturbation overhead, a more rigorous analysis would compare the models under identical batch sizes and potentially discuss whether AAT significantly increases VRAM consumption—a metric of equal importance to training time in industrial deployments.

3. Generalization Constraints of Handcrafted Perceptual Features: The reliance on traditional handcrafted feature extraction (SSIM, LBP, etc.) presents an inherent limitation to the model’s long-term adaptability. Although this approach yields impressive inference speeds, these analytic features often demonstrate limited generalization across diverse or novel degradation types. As noted by the authors, the model’s performance drops when encountering specific distortions such as radial geometric transforms. This indicates that the fixed feature set cannot fully encapsulate the vast complexity of human visual perception across all possible artifacts, particularly those underrepresented in legacy datasets. Furthermore, because the framework is structurally rigid based on these specific features, it may struggle to adapt to emerging distortion models (e.g., those generated by novel AI-based restoration) through standard evolutionary learning compared to more flexible, purely data-driven architectures.

---

> ### Author Rebuttal · Authors · 2026-03-28
>
> We thank the reviewer for the careful reading and constructive feedback. Although the reviewer noted rigorous theoretical and experimental contributions, we agree that the current manuscript should be clearer in several places and we will improve it in the revised version of the paper.
>
> **Training time, batch size, and VRAM.**
> We agree that the current Table 3 is not sufficiently described. The reported AAT-BiRQA number should not be interpreted as the time of adversarial fine-tuning alone: in our training pipeline, **AAT is applied as a fine-tuning stage after standard BiRQA training**, so the reported number corresponds to the full two-stage pipeline. We also agree that comparing BiRQA at a batch size of 32 against AAT-BiRQA at a batch size of 16 is not the right way to present training cost. We will correct this in the revision and report the matched-batch results along with peak VRAM usage.
> Under the same hardware setting, a controlled comparison at batch size 16 is:
> | Method / stage          | Batch size | Time (min) | Peak VRAM (GB) |
> | ---------- | ---------: | ---------: | -------------: |
> | BiRQA training          |         16 |        197 |            ~21 |
> | AAT fine-tuning only    |         16 |        154 |            ~32 |
> | AAT-BiRQA full pipeline |         16 |        351 |            ~32 |
>
> **Q1.** Yes. At the same batch size, AAT increases VRAM substantially. This is expected because adversarial fine-tuning introduces additional gradient-based perturbation computations within the training loop, increasing memory usage relative to standard clean training.
>
> **Q2.** Batch size 16 was chosen as a practical operating point for adversarial fine-tuning: it fit comfortably within the memory budget while still providing enough anchor/non-anchor comparisons within a batch. The method is not tied to this exact value, as we show in the additional ablation described below.
>
> **Q3.** We further evaluated AAT with batch sizes of 8, 16, and 32. When the anchor ratio was held constant, predictive performance and robustness remained largely stable, with corresponding SROCC values of 0.976, 0.978, and 0.977 on LIVE dataset and the same $R_{score}$ values. This suggests that AAT is largely insensitive to the absolute batch size, as long as the anchor coverage pattern is maintained. This is also consistent with the theory and our batch construction. For a fixed anchor ratio, the sampled MOS band width $R$ remains fixed, while a larger batch contains proportionally more anchors within that same band. As a result, the anchor spacing becomes denser, so the two-sided coverage term $\eta$ does not worsen and may decrease slightly. In this sense, increasing batch size with proportional anchor scaling preserves or mildly improves the theoretical coverage condition, while the main practical trade-off is still governed by the anchor ratio itself. The reported ratio was chosen to balance these two effects.
>
> **Generalization of handcrafted features.**
> Handcrafted perceptual features, indeed, despite offering computational efficiency, may be less flexible than fully learned representations when faced with novel or weakly represented distortion families. At the same time, we would like to clarify that the observed drop in geometric transformations does not appear to be specific to BiRQA alone.
> On **PieAPP** dataset, geometric transformations are the most difficult category for the compared IQA models, especially because these distortion types are **unseen during training**. In this category, all methods degrade relative to the other distortion groups, while BiRQA remains the strongest among the compared methods:
>
> | Method    |     Noise | Artifacts with regular patterns | Detail loss | Color change | Geometric transformations |
> | --------- | --------: | ------------------------------: | ----------: | -----------: | ------------------------: |
> | AHIQ      |     0.937 |                           0.851 |       0.821 |        0.840 |                     0.751 |
> | TOPIQ     |     0.944 |                           0.858 |       0.824 |        0.841 |                     0.778 |
> | BiRQA | 0.946 |                       0.862 |   0.825 |    0.844 |                 0.783 |
>
> We are grateful for the reviewer’s careful assessment. We hope that the clarifications provided here adequately address the concerns.

---

> > ### Author Rebuttal · Reviewer_pFv4 · 2026-04-04
> >
> > As the discussion stage is not yet over, I will refer to the opinions of other reviewers to give the final score

---

### Decision · Program_Chairs · 2026-04-30

**Decision:**

Accept (regular)

**Comment:**

This paper works on full-reference image quality assessment. Authors presented BiRQA, a compact FR IQA metric model that processes four fast complementary features within a bidirectional multiscale pyramid. Authors proposed anchored adversarial training to enhance robustness. Experimental results showed effectiveness of the proposed method.

The final rating of this paper are: 3 weak accept and 1 weak reject.

Before rebuttal reviewers mentioned

The strength of this paper are:
1) Effective Solution for Inference Speed and Adversarial Robustness. (Reviewer pFv4)
2) Rigorous Theoretical and Experimental Proof of Error Bounds. (Reviewer pFv4)
3) Comprehensive Experimental Validation and Rigorous Argumentation. (Reviewer pFv4)
4) identifies an important issue. (Reviewer iDsw)
5) anchored ranking formulation provides an alternative supervision signal. (Reviewer iDsw)
6) Robustness is an important property. (Reviewer iDsw)
7) addresses a practically important problem. (Reviewer 6NkS)
8) design is coherent and well motivated. (Reviewer 6NkS)
9) empirical study is reasonably solid. (Reviewer 6NkS)
10) efficiency results also make the method appealing for real-world deployment. (Reviewer 6NkS)
11) generally sound technical framework. (Reviewer fPkx)
12) paper is very writing. (Reviewer fPkx)
13) addresses two critical pain points in FR IQA. (Reviewer fPkx)

the weakness of this paper are:
1) Lack of Mathematical Clarity and Terminology Consistency. (Reviewer pFv4)
2) Non-Rigorous Methodology in Training Time Comparisons. (Reviewer pFv4)
3) Generalization Constraints of Handcrafted Perceptual Features. (Reviewer pFv4)
4) concerns on effectiveness of the anchored ranking formulation. (Reviewer iDsw)
5) does not provide evidence demonstrating how the improved robustness benefits real optimization tasks. (Reviewer iDsw)
6) novelty in model design is therefore somewhat limited. (Reviewer iDsw)
7) motivation behind selecting these specific descriptors is not clearly explained. (Reviewer iDsw)
8) methodological novelty appears somewhat incremental. (Reviewer 6NkS)
9) clean-performance gain over the strongest baselines is relatively modest.  (Reviewer 6NkS)
10) theoretical analysis mainly serves as a training justification rather than a strong robustness guarantee. (Reviewer 6NkS)
11) some technical claims lack sufficient support. (Reviewer fPkx)
12) unclear notation definitions in some formula. (Reviewer fPkx)
13) the significance is somewhat limited. (Reviewer fPkx)
14) The paper lacks truly novel core architectures or training paradigms. (Reviewer fPkx)

After rebuttal,
Reviewer pFv4, iDsw, 6NkS mentioned their concerns are fully addressed and maintained weak accept rating.

Reviewer fPkx still believed that methodological novelty is highly incremental and clean performance gain over SOTA is marginal, and thus maintained weak rejection score.

Given several reviewers thought the novelty of this paper is limited and many of them still thought this paper could be accepted given the strength of this paper, AC decided to given weak accept recommendation.